# Observational evidence for the formation of DMS-derived aerosols during Arctic phytoplankton blooms

Ki-Tae Park[1,†], Sehyun Jang[2,†], Kitack Lee[2,*], Young Jun Yoon[1,*], Min-Seob Kim[3], Kihong Park[4], Hee-Joo Cho[4], Jung-Ho Kang[1], Roberto Udisti[5], Bang-Yong Lee[1], Kyung-Hoon Shin[6]

[1]Korea Polar Research Institute, Incheon, 21990, Korea
[2]Division of Environmental Science and Engineering, Pohang University of Science and Technology, Pohang, 37673, Korea
[3]Environment Measurement & Analysis Center, National Institute of Environmental Research, Incheon, 22689, Korea
[4]School of Environmental Science and Engineering, Gwangju Institute of Science and Technology, Gwangju, 61005, Korea
[5]Department of Chemistry, University of Florence, Florence, 50016, Italy
[6]Department of Marine Sciences and Convergent Technology, Hanyang University, Ansan 1588, Korea

*Correspondence to*: Kitack Lee (ktl@postech.ac.kr), Young Jun Yoon (yjyoon@kopri.re.kr)

†These authors (Ki-Tae Park and Sehyun Jang) contributed equally

**Abstract.** The connection between marine biogenic dimethyl sulfide (DMS) and the formation of aerosol particles in the Arctic atmosphere was evaluated by analyzing atmospheric DMS mixing ratios, aerosol particle size distributions and aerosol chemical composition data that were concurrently collected at Ny-Ålesund, Svalbard (78.5° N, 11.8° E) during April and May 2015. Measurements of aerosol sulfur (S) compounds showed distinct patterns during periods of Arctic haze (April) and phytoplankton blooms (May). Specifically, during the phytoplankton bloom period the contribution of DMS-derived $SO_4^{2-}$ to the total aerosol $SO_4^{2-}$ increased by 7-fold compared with that during the proceeding Arctic haze period, and accounted for up to 70% of fine $SO_4^{2-}$ particles (< 2.5 µm in diameter). The results also showed that the formation of submicron $SO_4^{2-}$ aerosols was significantly associated with an increase in the atmospheric DMS mixing ratio. More importantly, two independent estimates of the formation of DMS-derived $SO_4^{2-}$ aerosols, calculated using the stable S isotope ratio and the non sea salt $SO_4^{2-}$/methanesulfonic acid ratio, respectively, were in close agreement, providing compelling evidence that the contribution of biogenic DMS to the formation of aerosol particles was substantial during the Arctic phytoplankton bloom period.

## 1 Introduction

Aerosols are known to have influenced the Earth's radiation budget, by scattering and absorbing incoming solar radiation or forming cloud condensation nuclei (CCN) (Charlson et al., 1992; Haywood and Boucher, 2000). However, large uncertainty remains in assessing the effects of aerosols on radiative forcing (IPCC 2013). Both field and laboratory studies have indicated that sulfate ($SO_4^{2-}$) is an aerosol component that principally contributes to the formation of CCN (Kulmala, 2003; Boy et al., 2005; Sipilä et al., 2010). Most $SO_4^{2-}$ particles originate from three sources: anthropogenic $SO_X$, sea salt $SO_4^{2-}$, and marine biogenic emissions (biogenic $SO_4^{2-}$). The latter is exclusively produced from the oxidation of dimethyl sulfide (DMS) (Simó, 2001).

Over the past 3 decades, DMS emissions from the upper ocean have been extensively studied because there may be a close linkage between marine biota and climate change (Charlson et al., 1987). DMS is of marine origin, is produced in the upper ocean via interactions of multiple biological processes (e.g., Kettle and Andreae, 2000; Stefels et al., 2007; Lee et al., 2012; Park et al., 2014), and some of the DMS is emitted into the atmosphere through air-sea gas exchange processes due to its low solubility in seawater. Atmospheric DMS is oxidized to methanesulfonic acid (MSA) and $H_2SO_4$, both of which influence the sulfurous aerosol budget (Davis et al., 1999; Barnes et al., 2006). The MSA and $H_2SO_4$ formed from DMS tend to transform into new particles via multiple nucleation processes (i.e., binary, ternary, and ion-induced) or condense onto existing particles because of their low volatility nature, and eventually form CCN (e.g., Korhonen et al., 1999; Vehkamäki et al., 2002; Lee et al., 2003). Several studies have investigated the association of DMS with non sea salt sulfate (nss-$SO_4^{2-}$), CCN, and aerosol optic depth (AOD) (e.g. Ayers and Gras, 1991; Gabric et al., 2005a; Carslaw et al., 2010), but direct proofs for such associations is limited (Ayers and Cainey, 2007). In the absence of this knowledge, in particular a lack of observational evidence for an association between DMS production and the formation and growth of aerosol particles in the marine boundary layer, doubts remain as to the validity of the hypothesized feedback mechanism connecting DMS-derived aerosols to climate (Quinn and Bates, 2011).

The occurrence of a large source of low volatility vapors (generally involving $H_2SO_4$) in the absence of coagulation with larger particles results in the constant formation of new particles in the atmosphere, because of the high level of production of condensable vapors and the relative lack of a condensation sink of pre-existing particles; these new particles subsequently grow into Aitken particles (> 50–100 nm in diameter), which probably influence cloud formation, and thereby radiation (Boy et al., 2005; Pierce et al., 2014; Leaitch et al., 2016). However, it is difficult to establish the quantitative relationship between oceanic DMS emission and the formation and growth of aerosol particles in the marine boundary layer. A small number of recent studies have reported that atmospheric DMS mixing ratios are related to the ocean phytoplankton biomass (Preunkert et al., 2008; Park et al., 2013), and that an increase in nss-$SO_4^{2-}$ corresponds to a proportional increase in the MSA concentration in regions of high phytoplankton productivity (Becagli et al., 2012, 2013 and 2016; Zhang et al., 2015), where DMS emissions are also high. However, a mechanistic understanding of the major physical and chemical processes that are involved in the formation of DMS-derived $SO_4^{2-}$ particles and their growth into larger particles remains elusive. Explicitly,

DMS emissions may exert greater impacts on aerosol formation in regions where the concentration of background aerosol particles is low, but DMS-producing phytoplankton are abundant. The Arctic atmosphere is an excellent example of an environment that meets these two criteria (e.g., Chang et al., 2011; Browse et al., 2012; Leaitch et al., 2013; Tunved et al., 2013; Willis et al., 2016).

The aims of the present study were to investigate the possible association between DMS emissions and the formation of aerosol particles, and to assess the contribution of DMS to the total $SO_4^{2-}$ aerosol budget. To this end we analyzed datasets of atmospheric DMS mixing ratio, aerosol particle size distributions, and aerosol chemical composition measured at Ny-Ålesund (Svalbard; 78.5° N, 11.8° E) in April and May 2015. To address the second aim we analyzed the MSA concentration (formed exclusively from the photo-oxidation of DMS) and the stable S isotope composition of aerosol particles.

**2 Experimental Methods**

The atmospheric DMS mixing ratio was measured at 1−2 h intervals on the Zeppelin observatory, which is located at an elevation of 474 m above sea level (m.a.s.l) and 2 km south and southwest of Ny-Ålesund. The measurement period (April – May) approximately covered the pre- to post-phytoplankton bloom periods. The analytical system includes a component for DMS trapping and elution and a gas chromatography (GC) equipped with a pulsed flame photometric detector (PFPD) enabling

DMS quantification. The detection limit of the analytical DMS system was reported to be 1.5 pptv in an air volume of approximately 6 L. Details for the DMS analytical system and the DMS measurement protocol are described in Jang et al (2016).

The distribution of aerosol particle sizes was measured at the Gruvebadet observatory, which is approximately 1 km southwest of Ny-Ålesund and approximately 60 m.a.s.l. Two discrete systems of scanning mobility particle sizer (SMPS) systems, each

of which included a differential mobility analyzer (DMA) and a condensation particle counter (CPC), continuously measured the distribution of small particles in in differential mobility equivalent diameter ranges 3−60 nm (combination of TSI 3085 and TSI 3776) and 10−500 nm (TSI 3034), and an aerodynamic particle sizer (APS) analyzed larger particles in the range 0.5−20 µm in diameter (Park et al., 2014; Lupi et al., 2016).

A high volume air sampler equipped with a $PM_{2.5}$ impactor (collecting particles < 2.5 µm in aerodynamic equivalent diameter)

was used for collection of aerosol samples. The sampler was mounted on the roof of the Gruvebadet observatory, and sampled particles every 3 days between 9 April and 20 May 2015, and later measured concentrations of major ions and the stable S isotope composition on a quartz filter.

For measurement of major ions ($Na^+$, $K^+$, $Mg^{2+}$, $SO_4^{2-}$, $Cl^-$, and MSA), a 47-mm (diameter) disk filter was punched out from a $PM_{2.5}$ aerosol quartz filter. All major ions collected on the disk filter were extracted in 50 mL Milli-Q water and analyzed by

a Dionex ion chromatography system (Thermo Fisher Scientific Inc., USA). The concentrations of major anions were determined using a Dionex model ICS-2000 with an IonPac AS 15 column and the concentrations of major cations were

determined using a Dionex model ICS-2100 with an IonPac CS 12A column. Three times the standard deviations of blank measurements were used as detection limits (0.01 to 0.26 ng mL$^{-1}$) (Kang et al., 2015).

For measurement of the stable S isotope ratio ($^{34}$S/$^{32}$S), all S compounds on half of a PM$_{2.5}$ quartz filter were extracted in 50 mL Milli-Q water. The filtrate was treated with 50–100 µL of 1M HCl to adjust the solution to pH = 3–4. Then, 100 µL of 1M BaCl$_2$ was added to cause all S as SO$_4^{2-}$ to precipitate as BaSO$_4$. After a 24-h precipitation period at room temperature, the BaSO$_4$ precipitate was recovered by filtration on a membrane filter and finally dried for 24 h. Each membrane filter containing BaSO$_4$ was packed into a tin cup and analyzed by isotope ratio mass spectrometer (IsoPrime100, IsoPrime Ltd, UK) coupled to an elemental analyzer (Vario EL, Elementar Co, German). The resulting S isotope ratio of a sample ($\delta^{34}$S) was expressed as parts per thousand (‰) relative to the $^{34}$S/$^{32}$S ratio in a standard (Vienna Diablo Troilite) (Krouse and Grinenko, 1991).

$$\delta^{34}S\ (‰) = \{(^{34}S/^{32}S)\ _{sample} / (^{34}S/^{32}S)\ _{standard} - 1\} \times 1000 \tag{1}$$

Information about the S isotope ratio of aerosol particles and the concentrations of major ions enabled the estimation of the relative contributions of biogenic DMS ($f_{bio}$), anthropogenic SO$_X$ ($f_{anth}$), and sea-salt SO$_4^{2-}$ ($f_{ss}$) to the total aerosol SO$_4^{2-}$ concentration. The concentration of ss-SO$_4^{2-}$ was first calculated by multiplying the Na$^+$ concentration (as a sea spray marker) by 0.252 (the seawater ratio of SO$_4^{2-}$/Na$^+$) (Keene et al., 1986). The nss-SO$_4^{2-}$ fraction of the total SO$_4^{2-}$ was then calculated by subtracting the fraction of ss-SO$_4^{2-}$ from the total SO$_4^{2-}$. Finally, the fraction of biogenic SO$_4^{2-}$ was estimated by solving the following equations:

$$\delta^{34}S_{measured} = \delta^{34}S_{bio}\ f_{bio} + \delta^{34}S_{anth}\ f_{anth} + \delta^{34}S_{ss}\ f_{ss} \tag{2}$$

$$f_{bio} + f_{anth} + f_{ss} = 1 \tag{3}$$

$$f_{ss} = 0.252\ Na^+/total\ SO_4^{2-} \tag{4}$$

In solving Eq. (2)–(4), the published S isotope end-member values of DMS-derived SO$_4^{2-}$ ($\delta^{34}S_{bio}$ = 18 ± 2‰), anthropogenic SO$_4^{2-}$ ($\delta^{34}S_{anth}$ = 5 ± 1‰), and sea-salt SO$_4^{2-}$ ($\delta^{34}S_{ss}$ = 21.0 ± 0.1‰) were used (Norman et al., 1999; Böttcher et al., 2007; Lin et al., 2012).

## 3 Results and Discussion

### 3.1 Atmospheric DMS and aerosol particles

The atmospheric DMS mixing ratio measured at Zeppelin observatory changed abruptly (by several orders of magnitude) within a few days of measurement and occasionally reached a level of 400 pptv, particularly during phytoplankton bloom events (Fig. 1a). The monthly mean DMS mixing ratio for May (47 ± 91 pptv) was more than double the April mean (18 ± 18 pptv). The 3-day integrated concentrations of MSA were broadly consistent with the concentrations of DMS; the lowest

concentration ($< 50$ ng m$^{-3}$) occurred in April and the highest value (approximately 200 ng m$^{-3}$) occurred in May (Fig. 1a). The strong positive correlation between the MSA concentrations and the corresponding DMS mixing ratios ($r = 0.84$, $n = 14$, $P < 0.05$; Fig. 1c) supports the assumption that the photochemical oxidation of biogenic DMS was the major source of MSA in our study area. Variations in DMS explained approximately 70% of the observed variance in the MSA concentration; the remaining variance was probably associated with variations in MSA that formed elsewhere, and was subsequently advected to the measurement site, and also with variations in the efficiency of photochemical oxidation of DMS.

The concentration of aerosol particles in the 3–10 nm diameter range (a nucleation mode), which is an indicator of recent nucleation, occasionally exceeded 3000 cm$^{-3}$. These small particles formed more frequently in May than in April (blue line in Fig. 1a). The observed increase in nucleation mode particles coincided with high atmospheric DMS mixing ratio and MSA concentration. Therefore, the 3-day mean DMS mixing ratios and the MSA concentrations were both significantly correlated with the 3-day mean concentration of nucleation mode particles ($r = 0.66$, $n = 14$, $P < 0.05$, Fig. 1d; and $r = 0.71$, $n = 14$, $P < 0.05$, Fig. 1e, respectively). Approximately 45% of the variability in the 3-day mean concentrations of nucleation mode particles can be explained by overall variations in the concentrations of DMS and MSA; some of the remaining variability will be associated with variations in the intensity of solar radiation, which influences the efficiency of photochemical oxidation of DMS. In fact, high atmospheric DMS mixing ratios found in mid-April ($77.1 \pm 51.5$ pptv; 14–17 April) was not followed by the formation of nucleation mode particles ($42.6 \pm 49.5$ cm$^{-3}$) and MSA (21.4 ng m$^{-3}$), possibly due to the low intensity of solar irradiance ($80.4 \pm 81.9$ W m$^{-2}$) (Fig. 1a, c and d). We cannot completely rule out the possibility that sources other than DMS contributed to the formation of nucleation mode particles (Fig. 1a, d and e). The emission of iodine is an alternative explanation for the particle nucleation event in the Arctic atmosphere during our study period (O'Dowd et al., 2002; Allan et al., 2015). Recent field observations in an iodine-rich coastal environment have shown that species containing iodine contribute to the formation of new aerosol particles via direct molecular-scale observations of nucleation in an iodine-rich coastal environment (Sipilä et al., 2016). As all chemical species (including H$_2$SO$_4$, iodine species, and organic vapors) that are directly involved in the nucleation process were not measured during the observational periods of the present study, we are unable to pinpoint the major contributor; however, these strong correlations between DMS and small aerosol particles indicate that these newly-formed particles were probably derived from recently released biogenic DMS.

## 3.2 Aerosol particles formed during periods of Arctic haze (April) and phytoplankton blooms (May)

Arctic haze, formed originally from emissions of pollution in North Europe, Siberia and North America, and its transport to the Arctic environment, has been reported to influence the aerosol characteristics of the Arctic atmosphere during early spring (April and earlier periods). The Arctic haze is a mixture of SO$_4^{2-}$ and particulate organic matter, plus minor contributions of ammonium, nitrate, dust, black carbon, and heavy metals (Quinn et al., 2007). The concentrations of nss-SO$_4^{2-}$ during this period reached 2000 ng m$^{-3}$, and the mean level in April was 2-fold greater than that in May (Fig. 2a). However, information about the concentrations of nss-SO$_4^{2-}$ only did not enable differentiation of the strengths of two major sources (anthropogenic

SO$_X$ vs. biogenic DMS). Additional measurements of particle concentrations enabled quantification of the contributions of anthropogenic SO$_X$ and biogenic DMS to the total nss-SO$_4^{2-}$.

The transition of aerosol microphysical properties from a distribution dominated by an accumulation mode (Arctic haze period) to a distribution dominated by nucleation and Aitken mode atmospheric particles (phytoplankton bloom period) was probably driven by the combination of three factors, including changes in air mass transport, incoming solar radiation and condensation sink processes (Tunved et al., 2004 and 2013). Specifically, the large accumulation mode particles outnumbered the small nucleation and Aitken mode particles during early spring (April) but the concentration of those large particles decreased rapidly from April to May, with particles smaller 100 nm becoming dominant in May (Fig. 3 and S1). A similar sharp transition (large-to-small particles) in the dominant particle type was also identified in previous observations at the same site (Engvall et al., 2008).

Our data on the particle size distributions showed that particles > 100 nm were more abundant in April, whereas small particles (< 100 nm) were more abundant in May (Fig. 3 and S1). As a result, the total surface area of aerosol particles in April was 2-fold greater than that observed in May, whereas the concentration of particles in April was 3-fold less than that in May (Fig. 3). As the condensation sink is proportional to the surface area of aerosol particles, it will decrease with decreasing intensity of Arctic haze. The concentrations of nss-SO$_4^{2-}$ measured in April did not correlate with the levels of biogenic MSA (P > 0.05; blue circles in Fig. 2b). On the contrary, a strong correlation ($r = 0.75$, $n = 7$, P < 0.05; red circles in Fig. 2b) between these two parameters was found in May. The greater DMS contribution to the formation of nss-SO$_4^{2-}$ in May than in April is broadly consistent with the 2-fold greater chlorophyll concentration observed in May compared to April (Fig. S2). These observations support that the formation of new particles resulting from the photo-oxidation of biogenic DMS, followed by a gas-to-particle conversion is an important source of Aitken mode (10–100 nm) particles in the Arctic atmosphere. Recent field observations also provided the evidence that the growth of nucleation mode particles in the Arctic summer atmosphere can be mediated by the presence of secondary marine organic aerosols, including MSA (Willis et al., 2016).

## 3.3 SO$_4^{2-}$ aerosol particles formed from biogenic DMS

The use of an asymptotic value in a plot of nss-SO$_4^{2-}$/MSA ratio versus MSA concentration is a convenient method for estimating the fraction of biogenic SO$_4^{2-}$ aerosols. As the MSA concentration in an aerosol sample increases (Fig. 4), the contribution of biogenic SO$_4^{2-}$ to the total nss-SO$_4^{2-}$ will also increase while the contributions of other sources will decrease. In this case, the nss-SO$_4^{2-}$/MSA ratio tends to approach an asymptotic value as the MSA concentration increases. Therefore, this asymptotic value adequately represents the nss-SO$_4^{2-}$/MSA ratio derived exclusively from DMS (Udisti et al., 2012, 2016). The biogenic SO$_4^{2-}$/MSA ratio has been reported to vary considerably in space and time (Gondwe et al., 2004), because the ratio is sensitive to temperature, and to a lesser extent photochemical species or reactions with halogen radicals (Bates et al., 1992). When this method was applied to data for aerosol samples (PM$_{10}$) collected in 2014 at the Gruvebadet observatory, in a vicinity of our DMS measurement site, the biogenic SO$_4^{2-}$/MSA ratio was estimated to be 3.0 (Udisti et al., 2016) (Fig. 4). In other polar locations, a ratio of 2.6 was reported, including for Alert station (82.5$^o$ N, 62.3$^o$ W; 210 m.a.s.l.) (Norman et al.,

1999) and Concordia station (75.1° S, 123.3° E; 3233 m.a.s.l) (Udisti et al., 2012). We estimated the amount of biogenic $SO_4^{2-}$ by multiplying the biogenic $SO_4^{2-}$/MSA ratio (3.0) by the MSA concentration in each aerosol sample. The fraction of anthropogenic $SO_4^{2-}$ was estimated by subtracting the combined ss-$SO_4^{2-}$ plus biogenic $SO_4^{2-}$ concentration from the total $SO_4^{2-}$ concentration.

Another method for estimating biogenic $SO_4^{2-}$ is to use S isotope ratios ($\delta^{34}S$) of $SO_4^{2-}$ aerosols, because the $\delta^{34}S$ values of biogenic DMS ($18 \pm 2‰$) are greater than those of anthropogenic $SO_4^{2-}$ ($5 \pm 1‰$) but less than that of sea salt ($21.0 \pm 0.1‰$) (e.g., Wadleigh, 2004; Lin et al., 2012, Oduro et al., 2012). A wide range in $\delta^{34}S$ (0–8‰) has been reported for anthropogenic $SO_2$ compared with values reported for other sources (Krouse and Grinenko, 1991). Surprisingly, Patris et al (2000) reported consistent regional-scale $\delta^{34}S$ values for anthropogenic $SO_2$. For example, in remote Arctic regions (including Ny-Ålesund

and Alert) the S isotope ratios measured for $SO_4^{2-}$ aerosols during the Arctic haze period in a single year mostly fell within the narrow range of 5–6‰ (McArdle and Liss, 1995; Norman et al., 1999), probably because regional-scale mixing processes averaged the signals (Partis et al., 2000). During the study period the $\delta^{34}S$ values measured at Ny-Ålesund ranged from 4.6 to 10.3. The $\delta^{34}S$ values were higher in May (8.8–10.3‰) than in April (4.6–8.2‰), reflecting changes in S sources.

The contributions of anthropogenic and biogenic $SO_4^{2-}$ to the total $SO_4^{2-}$ aerosols, estimated using two independent methods,

are shown in Fig. 5 (and Fig. S3). In April and May the contribution of ss-$SO_4^{2-}$ to total $SO_4^{2-}$ was small (< 3% of total aerosol particles < 2.5 µm in diameter). It was estimated that approximately 90% of the total $SO_4^{2-}$ was of anthropogenic origin in April, when the Arctic haze was most intense. This estimation is consistent with measurements of anthropogenic $SO_4^{2-}$ in $PM_{10}$ aerosols collected in April 2014 at the same site (Udisti et al., 2016). In May, following the period of Arctic haze, the contribution of DMS-derived $SO_4^{2-}$ sharply increased up to 70% of fine $SO_4^{2-}$ particles (< 2.5 µm in diameter), and this

corresponded to the Arctic phytoplankton bloom (Fig. 5a and S3). The concentrations of DMS-derived $SO_4^{2-}$ estimated using the MSA-based approach were approximately 50% higher than the values based on the S-isotope method. The differences may be the result of uncertainties associated with uncertainties in assigned S isotope end-member values, and the ratio of biogenic $SO_4^{2-}$ to MSA. Nonetheless, a robust correlation between these two estimates was found during April and May ($r = 0.74$, $n = 13$, $P < 0.05$; Fig. 5b).

**3.4 Relationship between concentrations of DMS-derived $SO_4^{2-}$ and the concentrations of aerosol particles**

In both April and May, the concentrations of biogenic $SO_4^{2-}$ particles estimated using the MSA-based and S-isotope-based methods were significantly correlated with the concentration of small aerosol particles in nucleation (from 3–10 nm, $r = 0.71$, $n = 14$, $P < 0.05$; Fig. 6a) and Aitken modes (from 10–100 nm, $r = 0.89$, $n = 14$, $P < 0.05$; Fig. 6b). However, when the Arctic haze prevailed in April, the concentration of large particles in accumulation mode (from 100 nm − 1 µm, $P > 0.05$; Fig. 6c)

was not significantly correlated with the concentration of biogenic $SO_4^{2-}$, but was strongly correlated with the concentration of anthropogenic $SO_4^{2-}$ ($r = 0.92$, $n = 7$, $P < 0.05$; the inset in Fig. 6c). In contrast, when Arctic haze moderated in May and the abundance of phytoplankton began to increase, the concentration of biogenic $SO_4^{2-}$ was strongly correlated with the concentration of accumulation mode particles ($r = 0.91$, $n = 7$, $P < 0.05$; Fig. 6c). A strong correlation between biogenic $SO_4^{2-}$

particles and the surface areas of particles provided additional evidence that biogenic $SO_4^{2-}$ significantly contributed to small particle formation (Fig. S4). Moreover, the formation and growth of sub-micrometer particles with diameters between 3 and 100 nm coincided with high concentrations of biogenic $SO_4^{2-}$ (Fig. 7 and S5). It is noteworthy that the contribution of anthropogenic $SO_4^{2-}$ to fine $SO_4^{2-}$ particles (< 2.5 µm in diameter) was still considerable (30–60%) during the bloom period

(May) (Fig. 5a and S3). Therefore, in May anthropogenic $SO_4^{2-}$ may also have partly contributed to the formation and growth of small aerosol particles (< 100 nm). However, we could not accurately estimate the exact contributions of biogenic versus anthropogenic $SO_4^{2-}$ to the formation and growth of aerosol particles in the absence of information on the chemical composition of size-segregated aerosol particles. The chemical composition of size-segregated aerosol particles was recently measured in the Arctic atmosphere during summer months. More than 60% of the aerosol particles havoing a diameter < 0.49 µm was

found to be derived from biogenic $SO_4^{2-}$ (Ghahremaninezhad et al., 2016). According to a study based on an aerosol microphysics box model (Chang et al., 2011), the atmospheric DMS mixing ratios observed during phytoplankton bloom periods in our study were sufficiently high for the formation of ultrafine aerosol particles, when background particle concentrations are low (i.e. DMS mixing ratio > 100 pptv; condensation sink < 7.0 $m^{-2}$) (Fig. 1a and b).

These direct observations provide evidence that biogenic DMS released from the Arctic Ocean contributed to the formation of

new aerosol particles, and their subsequent growth to larger climate-relevant particles (> 50–100 nm in diameter). In particular, new particles appeared to be formed more rapidly in May, and grew more efficiently to large climate-relevant particles, possibly because of the presence of a large source of condensable vapor (generally involving $H_2SO_4$) formed from the oxidation product of DMS and the low concentration of background aerosol particles (> 100 nm). During phytoplankton bloom events, the Arctic environment surrounding the DMS observation site was found to have three key characteristics (small surface area

of particles, high concentrations of low volatility condensable vapors, and high solar radiance) that can greatly facilitate the formation of new particles from biogenic DMS.

## 4 Conclusion

The Arctic Ocean is warming faster than any other ocean region (IPCC 2013). Consequently, a corresponding decrease in the sea ice extent may impact on primary production and the sea-to-air flux of climate-relevant gases including $CO_2$ and DMS

(Gabric et al., 2005b). The loss of Arctic sea ice may also result in a new source of aerosol particles and CCN from DMS, possibly counterbalancing the decrease in surface albedo by an increase in cloud albedo (Levasseur, 2013; Woodhouse et al., 2013).

This study demonstrated the close association between an increase in DMS and increases in the total mass concentration of nss-$SO_4^{2-}$ and MSA during the period of Arctic phytoplankton blooms. It also confirmed that the increase in DMS-derived

$SO_4^{2-}$ occurred concurrently with the formation and growth of aerosol particles. Measurements of the atmospheric DMS mixing ratio, MSA, S isotope ratio, aerosol particle size distributions, and satellite-based biomass indicated that there was a connection between oceanic DMS emissions and the formation of aerosol particles in the Arctic atmosphere during the phytoplankton

bloom period. Further measurements of the chemical composition of marine aerosols less than 100 nm in size are needed to provide more direct evidence for the contribution of key nucleating compounds (i.e., DMS, iodine-containing species, and organic vapors) to the formation of aerosol particles, and to investigate how aerosol particles in the Arctic are responding to climate change.

5 **Author contribution**

K.-T.P., K.L, Y.J.Y and B.-Y.L designed the study. K.-T.P and S.J carried out the field campaign for DMS and aerosol measurements. M.-S.K, K.-H.S analyzed S-isotope. K.P, H.-J.C and R.U analyzed aerosol particle size distribution. J.-H.K analyzed major ions including MSA. K.-T.P., K.L and Y.J.Y wrote the manuscript.

**Acknowledgement**

10 This research was supported funded by the National Research Foundation (NRF) of Ministry of Science, ICT and Future Planning (Global Research Project; Mid-career Researcher Program, No. 2015R1A2A1A05001847; NRF-2016M1A5A1901769) and KOPRI-PN17081 (CAPEC project; NRF- 2016M1A5A1901769). Partial support was provided by Management of Marine Organisms causing Ecological Disturbance and Harmful Effects" funded by the Ministry of Oceans and Fisheries and KOPRI-PN17140. Solar irradiance data were provided by the Alfred Wegener Institute, Helmholtz Centre 15 for Polar and Marine Research (doi: 10.1594/PANGAEA.873812).

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

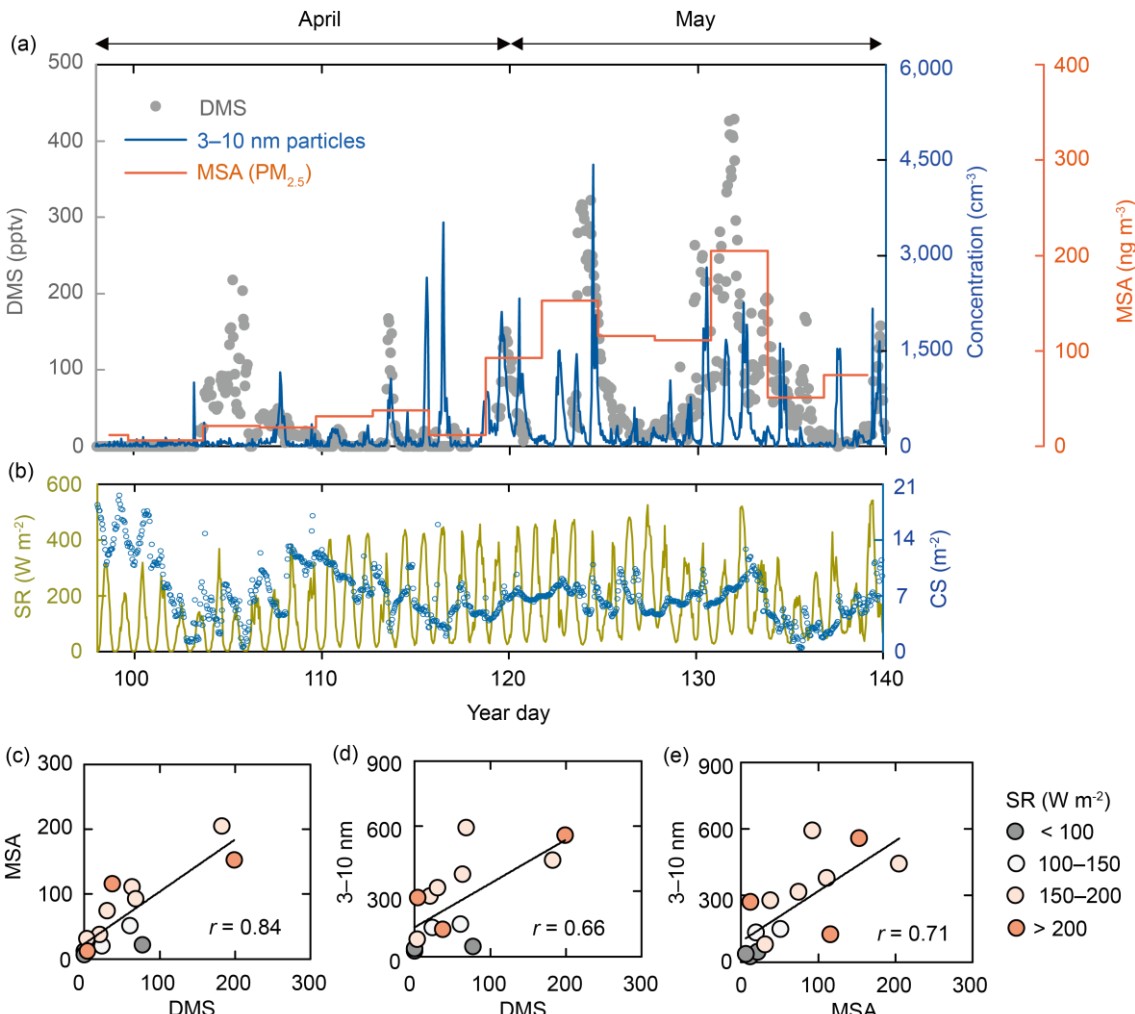

**Figure 1:** (a) The mixing ratios of atmospheric DMS measured at the Zeppelin observatory (gray circles), the concentration of nucleation mode particles (3–10 nm in diameter) measured at the Gruvebadet observatory (1 km distant from Zeppelin station) (blue line) and the concentration of MSA collected at the observatory (red line) during April and May 2015. (b) The solar irradiance (SR) measured in Ny-Ålesund (yellow line), and the condensation sink (CS) calculated by Kerminen et al. (2004) (blue circles). (c) The relationship between the atmospheric DMS mixing ratio, averaged for 3 days, and the corresponding MSA concentration. (d) The relationship between the atmospheric DMS mixing ratio, averaged for 3 days, and the corresponding concentration of nucleation mode particles. (e) The relationship between the MSA concentration and the corresponding concentration of nucleation mode particles. The colors of the circles in Fig. 1c–1e indicate the intensity of solar irradiance (SR; integrated over a wavelength range of 200–3600 nm).

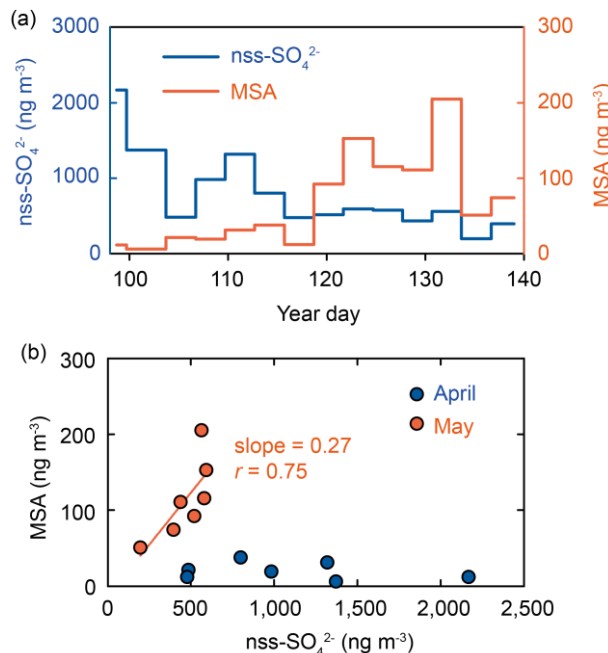

**Figure 2:** (a) The concentrations of nss-SO$_4^{2-}$ (blue line), and MSA (red line) at the Gruvebadet observatory in April and May 2015. (b) The relationship between nss-SO$_4^{2-}$ and MSA measured in April (blue circles; Arctic haze period) and in May (red circles; phytoplankton bloom period). The red solid line shown in Fig. 2b indicates the best fit.

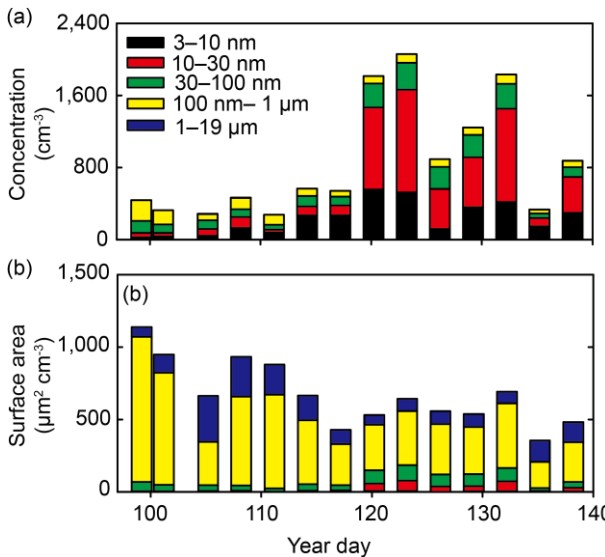

**Figure 3:** (a) Concentration, and (b) surface area of aerosol particles, including nucleation mode (3–10 nm), Aitken mode (10–100 nm), accumulation mode (100 nm– 1 µm) and coarse mode (1–19 µm) particles in April and May 2015.

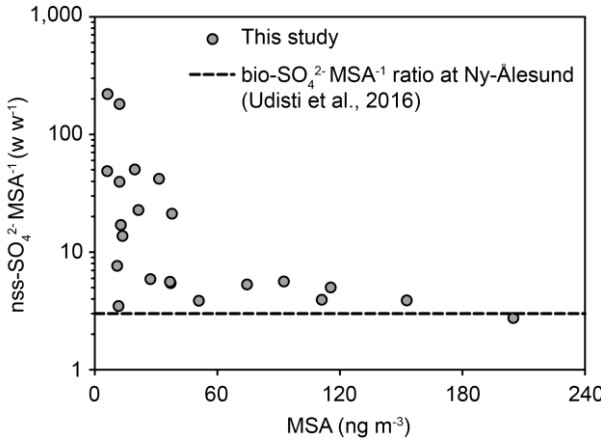

**Figure 4:** Relation between the nss-SO$_4^{2-}$/MSA ratio and the MSA concentration (PM$_{2.5}$; April and May 2015; grey circles).
5   The dashed line indicates the bio-SO$_4^{2-}$/MSA ratio measured at Ny-Ålesund in 2014 (Udisti et al., 2016).

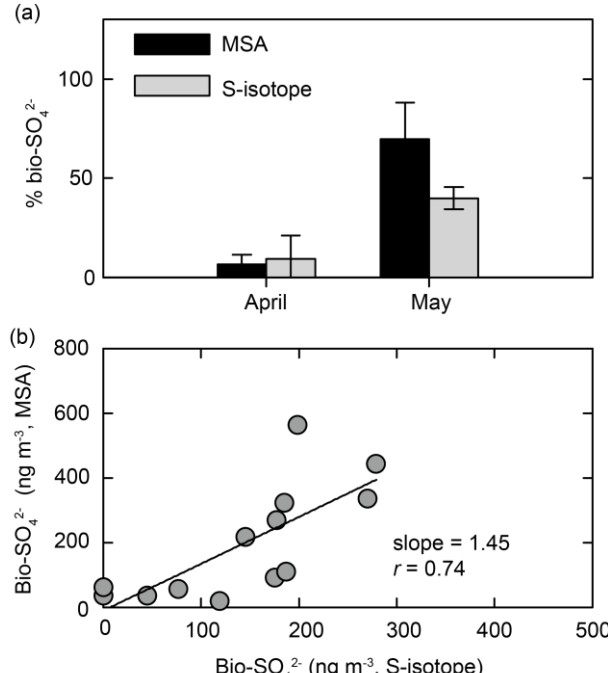

**Figure 5:** Biogenic (bio) $SO_4^{2-}$ estimated using MSA and S isotope. (a) Biogenic $SO_4^{2-}$ as a percentage of the total aerosol $SO_4^{2-}$ burden. Black bars: biogenic $SO_4^{2-}$ estimated using MSA; grey bars: biogenic $SO_4^{2-}$ estimated using stable S isotope. (b) The relationship between the concentrations of biogenic $SO_4^{2-}$ estimated using MSA, and the concentrations of biogenic $SO_4^{2-}$ estimated using stable S isotope. The black solid line represents the best fit.

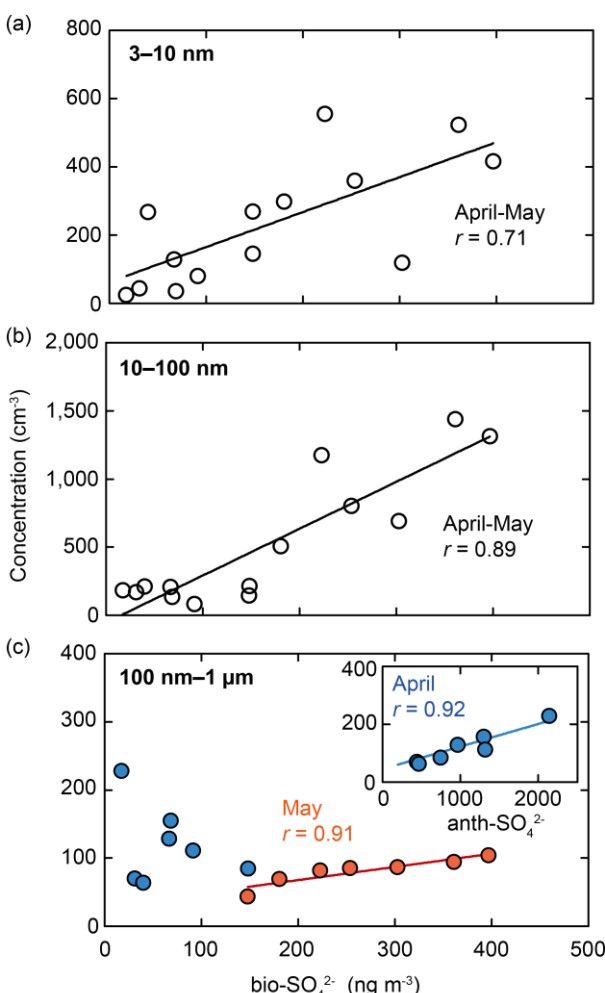

**Figure 6:** Relationships between the concentration of biogenic (bio) $SO_4^{2-}$ and the concentrations of particles of various sizes in April and May. (a) Particles 3–10 nm, (b) particles 10–100 nm, and (c) particles 100 nm–1 μm. The black solid lines in Fig. 6a and 6b indicate the best fit between biogenic $SO_4^{2-}$ and the particle concentration during April and May 2015. Blue and red circles in Fig. 6c indicate data obtained in April (Arctic haze period) and May (phytoplankton bloom period), respectively. The red solid line in Fig. 6c indicates the best fit between biogenic $SO_4^{2-}$ and particle concentration in May 2015, and the blue solid line in the inset of Fig. 6c indicates the best fit between anthropogenic (anth) $SO_4^{2-}$ and particle concentration in April 2015.

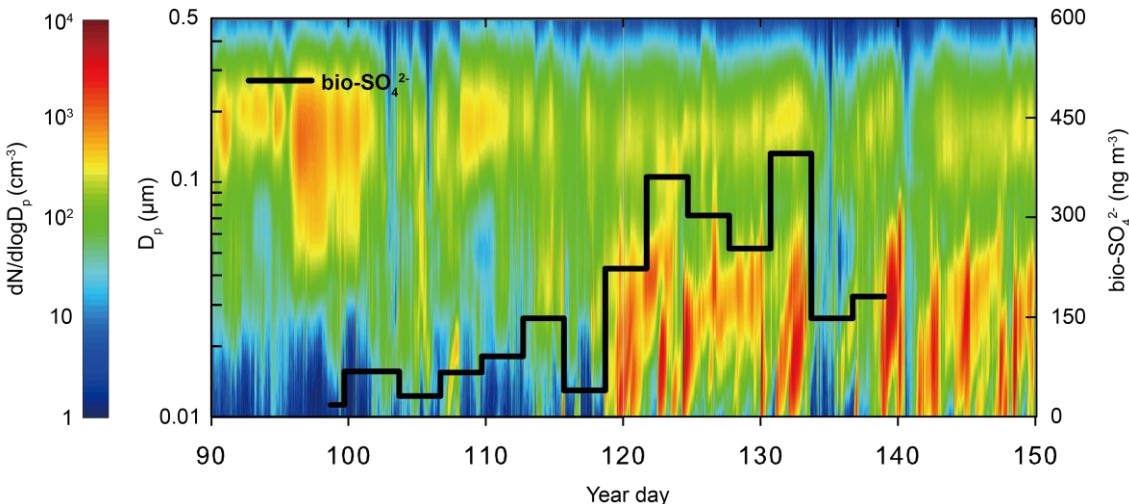

**Figure 7:** Spectral plot of number size distribution ($dN/dlogD_p$) as a function of particle diameter ($D_p$, 10–500 nm) and year day during April and May 2015. The black line represents the concentration of biogenic $SO_4^{2-}$.