# Peer review of "Observational evidence for the formation of DMS-derived aerosols during Arctic phytoplankton blooms"

_Atmospheric Chemistry and Physics, 2016_

## Referee Comment (RC1) · Anonymous Referee #1 · 22 Feb 2017

General – The authors present a unique combination of observations at Ny-Ålesund, Svalbard made during April and May of 2015 that are used to connect biogenic sulphate (from DMS oxidation) to new particle formation (NPF). I find the connection reasonably compelling. The paper is well structured and mostly well written, but I have technical questions about some of the discussions that need to be addressed. As well, there are recent relevant references that should be considered. The subject is appropriate for ACP, and it should be published if the authors attend to some details of the discussions in various areas of the paper.

Comments:

1. Check the references. Some referred to in the text are not in the list.

[Figure]

2. Page 2, line 12 - "DMS is emitted..."

3. Page 2, line 13 – remove "rapidly" or quantify it.

4. Page 2, line 15 - The attraction between H2SO4 and water is an important factor in particle nucleation, and so in a broad sense there is a connection between Henry's Law constant and the ability of H2SO4 to contribute to particle nucleation, but Henry's law constants are macroscopic representations while particle nucleation is a molecular-level process. Are you saying that the Henry's Law constants control particle nucleation, or are you mostly referring to what factors might control the condensation of H2SO4 and MSA onto existing particles? Please elaborate, and also note that condensation may occur in the absence of water on a particle.

5. Page 2, lines 21-22 – The use of condensation here in the context of new particle formation contradicts your sentence above that states that MSA and H2SO4 are either transformed into new particles or condense on existing particles. Please correct.

6. Page 2, line 23 – For a reference to these small particles influencing Arctic cloud, see Leaitch et al. (ACP, 2016).

7. Page 2, lines 29-33 – Many references exist that are relevant to the discussion here. For example: Engvall et al. (ACP, 2008); Chang et al. (JGR, doi:10.1029/2011JD015926, 2011); Browse et al (ACP, 2012); Leaitch et al., Elementa (2013); Tunved et al. (ACP, 2013); Willis et al. (ACP, 2016); Croft et al. (Nat. Comm., 2017).

8. Page 3, Lines 15-19 – Was one DMA and CPC used to measure both indicated size ranges, or were there two DMA-CPC systems? Please provide either a reference for these measurements or the details of sampling and calibration.

9. Page 4, line 21 – The delta 34S value for sea salt is expressed without uncertainty, whereas the deltas for anthropogenic and biogenic S have uncertainties (or ranges). Please elaborate.

10. Page 4, lines 30-32 – Are you trying to say that local oxidation of DMS was responsible for most of the MSA as opposed to more distant production of MSA? Figure 1b suggests that about 70% of the variance in MSA was connected directly with your measured DMS, but some MSA may still have been advected into the area. E.g. MSA levels below about 30-40 ng m-3 do not appear to be strongly associated with DMS. Please provide a more detailed and clearer discussion.

11. Figure 1 – This figure is impressive. It would be helpful to have the time series of solar irradiance in 1a or as a separate panel, even with SR represented in 1b, 1c and 1d. It would also help with your discussion on lines 9-11 of page 5.

12. Page 5, line 10 – "possibly" rather than "probably".

13. Page 5, line 11 – add a reference for iodine (e.g O'Dowd, Nature or Allan, ACP).

14. Page 5, lines 24-28 – I have a few little concerns about these sentences. It is unclear why you suggest that neutralization especially enhances growth into the accumulation mode. Acidic sulphate is hygroscopic, just as the salts; slightly more so. Are you trying to say that very acidic solutions will inhibit further transfer of acids from the gas phase? Sulphate accumulates in particles >100 nm through condensation of H2SO4 from the gas-phase, coagulation and through aqueous-phase production of H2SO4 (e.g. in cloud droplets). Neutralization of acidic sulphate may play a significant role in these processes, but NH3 may also be an important factor in the production of new particles in the Arctic (e.g. Croft et al., Nat. Comm., 2017). You indicate that the sulphate from DMS oxidation gives rise to smaller particles, but particle production and evolution doesn't care whether the H2SO4 is anthropogenic or biogenic. In a situation of relatively low gaseous precursor concentrations (as can be the case in the Arctic and from DMS), low existing particle surface area will help nucleation. Available sunlight will be a factor in the production of gas-phase H2SO4, but H2SO4 from biogenic sources is the same as H2SO4 from anthropogenic sources. The discussion here needs improvement. Also, remove "established explanations".

15. Page 5, line 32 – Sharma et al did not present particle number concentrations.

16. Page 6, lines 3-10 – This is particularly interesting because the decreases in the number concentrations and surface areas of the 100-1000 nm particles that are associated with the increase in sub-30 nm particles from April to May appear to be much lower than the decrease needed for NPF at Alert (Leaitch et al., Elementa, 2013). Possibly one reason is the proximity to the sources: open water is close by Svalbard year round with potential for higher concentrations of SO2 from DMS, whereas Alert is more distant from large areas of open water. Willis et al (ACP, 2016) document a case of NPF and growth associated with open water in the summertime Arctic that may be conceptually similar to your work.

17. Page 6, lines 10-12 - Reference is made to the Henry's Law constants being responsible for gas-to-particle nucleation. Please elaborate, considering comment 4 above.

18. Page 6, Fig. S3 – This figure seems to be a fundamental part of the discussion, and I believe it belongs in the main text rather than a supplement. This figure also demonstrates that some lower concentrations of MSA may result from distant transport to the site.

19. Page 6, line 29 – Why here do you give a range (0-8) for the deltaS of anthropogenic sulphate, when above (for equation 2) you give a value of 5(+/-1)? Please be consistent.

20. Page 7, Figures 4 and 5 – These are compelling results, but you need to acknowledge that there is still a significant fraction of the sulphate that is anthropogenic in origin, and that you cannot identify the origin of the H2SO4 contributing to the new particles. It is possible that the NPF is due to anthropogenic SO2 (or a combination of anthropogenic and biogenic SO2), and that the correlations result mostly from the coincidence of a lower condensation sink with the relative increase in biogenic S.

21. Page 8, line 9 – Please change "confirming" to "indicating" or "showing". There is no hypothesized relationship to confirm.

---

## Referee Comment (RC2) · Anonymous Referee #2 · 17 Apr 2017

Review of:

Observational evidence for the formation of ocean DMS-derived aerosols during Arctic phytoplankton blooms Ki-Tae Park, Sehyun Jang, Kitack Lee, Young Jun Yoon, Min-Seob Kim, Kihong Park, Hee-Joo Cho, Jung-Ho Kang, Roberto Udisti, Bang-Yong Lee, Kyung-Hoon Shin

General This is a good study that illustrates a correlation between marine productivity, gaseous emissions of biogenic sulphur, and new particle production in the Arctic region and contrasts a period dominated by marine emissions with that of long-range pollution transport leading to Arctic haze. The authors clam that this study proves a conclusive link, including direct evidence, of DMS emissions leading to new particle

production, and presents nss-sulphate as the predominant aerosol species of which, isotope analysis concludes the majority sulphates is marine biogenic. The authors overstate their results and their conclusions where they claim DMS-derived sulphate leads to the formation (presumably nucleation and/or growth of nucleated clusters into nucleation mode aerosols at 3-10 nm) of new particles when there are no measurements to distinguish between inorganic, organic or halogen species, the latter which has been shown to overwhelm sulphuric acid in terms of contribution to the particle formation process, even though sulphate may ultimately dominate total mass. In summary, it is a useful study but the results should not be overstated. If the authors wish to maintain their assentation that the new particles are formed by DMS emissions, they have to demonstrate that sufficient sulphuric acid concentrations for nucleation and cluster growth to a few nm can be achieved from the levels of DMS encountered and that the occurrence of the new particle peaks coincide with the peak production rate of sulphuric acid. Recommendation: accept with minor revisions to texts to be consistent with the limitations of the results or if to maintain the claims of what these results illustrate, provide model assessments of the concentration of sulphuric acid available for particle production via DMS oxidation. Specific Comments Title : 'remove ocean' from title. Abstract: The results also showed that a sharp increase in the atmospheric DMS mixing ratio during Arctic phytoplankton bloom events was directly associated with the formation of sub-micrometer SO4 aerosols, and their subsequent growth to climate-relevant particles.

I would state it the other way around – the formation of submicron sulphate aerosol was associated with increased atmospheric DMS mixing ratios. Introduction

Paragraph starting line 9. The text states that there may be a direct link between marine biota and climate change. I suggest this link is hardly direct. Later, line 14 onwards, states "in particular a lack of observational evidence for a direct association between DMS production and the formation and growth of aerosol particles.". the two statements are inconsistent.

Line 19-20: I am not sure that the validity of the DMS-climate feedback is in doubt, is it not more the case that has been illustrated that in some regions, this feedback has been shown to be not as significant as perhaps other possible feedbacks or not the only chemical species in the feedback loop. For example, iodine has been suggested to be more important in the actual nucleation processes (O'Dowd et al., Nature, 2002; Sipila et al., Nature, 2016); and organic vapours may dominate in the growth of clusters into stable aerosol particles as often there is estimated to be insufficient sulphuric acid to account for initial growth processes [Dall'Osto, M., et al. (2012), J. Geophys. Res., 117, D12311, doi:10.1029/2012JD017522.]. Moreover, the majority of sulphate formation is estimated to be via the aqueous-phase of heterogenous pathway rather than the homogenous pathway [Hoppel et al., JGR, 99, 14,443-14,459, 1994, ].

In addition, the role of primary aerosol feedback, in contrast to secondary aerosol, and involving organics with high activation efficiency [e.g. O'Dowd et al, Nature, 2004; Ovadnevaite et al, GRL, 2011; O'Dowd et al., Scientific Reports, 2015].

Experimental set up Page 3 Line 15: please be specific on the online aerosol size distribution measurements, neither a DMA or a CPC on their own measure size distributions. Were they combined in a SMPS, or DMPS, what configuration? Were the particle dry size, or partly hydrated? Very few SMPS/DMPS size distributions reported are 100% dry size – more is needed here.

Results and Discussion Page 4 line 27: 3 fold 18 is 54 not 47, I would rephrase to "more than double" Page 5 line 1. "The concentration of aerosol particles in the 3–10 nm diameter range (a nucleation mode), which is an indicator of recent nucleation, occasionally exceeded 3000 cm–3. These small particles formed more frequently in May than in April (blue line in Fig. 1a). As a result, the 3-day mean DMS mixing ratios and the MSA concentrations were both significantly correlated with the 3-day mean concentration of nucleation mode particles (r = 0.66, n = 14, P < 0.05, Fig. 1c; and r = 0.71, n = 14, P <0.05, Fig. 1d, respectively); the observed nucleation events also concurrently occurred with high 5 atmospheric DMS mixing ratios (Fig. 1a). This does

not make sense – this means that, as a result of the small particles being indicators of recent nucleation, and being formed more frequently in May rather than April, DMS and MSA was significantly correlated to nucleation mode particles. This is not scientifically correct.

Line 6: 45% of the variability in new particles can can be explained by DMS/MSA then line 11 states "We cannot completely rule out the possibility that sources other than DMS (e.g., iodine) contributed to the formation of nucleation mode particles (Fig. 1a, c and d). However, these strong correlations indicate that the small aerosol particles that were formed newly were probably derived from recently released biogenic DMS."

To quote from Sipila et al: "In Greenland, we began to observe elevated concentrations of $HIO_3$ after sunrise in late February, often associated with new particle formation events. During such events, the $HIO_3$ concentrations tended to be much higher than that of sulfuric acid (Supplementary Fig. 12), and it seems that the cluster formation could be explained almost entirely by the $HIO_3$ clustering mechanism." I think it has to be accepted and stated that there is no direct evidence in this study that can state DMS-derived species were responsible for the production of new particles given that $HIO_3$ was not measured as is also likely to be emitted (or more precisely, produced) in parallel to DMS (or DMS products).

The line 31 same page "The high concentration of small particles (< 100 nm) during phytoplankton bloom period (May) constitutes compelling evidence for new particle formation derived from local DMS emission" Probably derived and compelling evidence are not the same thing.

I do not think you can rule out iodine in cluster formation and the nucleation processes and even perhaps organics contributing to particle formation. We have no handle on whether or not sulphuric acid was present in sufficient concentrations to nucleate clusters or particularly grow clusters to measurable sizes. If you are going to make such strong statements you need to present solid arguments, for example even a box model

of the maximum concentrations of sulphuric acid likely to be achieved for the DMS concentrations, oxidation rates/solar radiation and condensation sinks. A number of groups have illustrated what concentrations are required for various nucleation and aerosol formation pathways, why not apply this approach here to present some more comprehensive arguments.

Page 6, line 4, I think these days, condensation sink is used in this context more than surface area. What role had supermicron particles in surface area or condensation sink?

Line 9-11 it is stated : These observations support our hypothesis that the formation of new particles resulting from the photo-oxidation of biogenic DMS . Where have the authors presented a hypothesis or tested a hypothesis? The authors have only made tenuous links to new particle formation and DMS. The observations do not support any such hypothesis in any event. If they want to present such hypothesis', and test it, the manuscript needs to be restructed. For example, the seemingly definitive case-closed statement on line 9-11 comes before the whole section on Aerosol formation from biogeic DMS (setion 3.3). BTW, is there non-biogenic DMS?

Section 3.3 Aerosol formation from biogeic DMS, while it illustrates a link between DMS and SO4 aerosol, this does not present anything on the link between new particle formation and DMS or SO4.

How much of the aerosol mass is accounted for by sulphate and MSA and how much sea salt and or organics. Is there closure between the SMPS/DMPS/APS derived mass distributions, or integrated mass distributions up to 2.5 microns and sulphate/seasalt.

Von Glassow and Crutzen found that in cloud free conditions les to 5-15% of DMS being converted to sulphate while under cloud conditions, 100% conversion occurs. How much of the DMS is available for nucleation/condensation of sulphuric acid to the aerosol phase and over what timescale?

Conclusions The conclusion is poorly presented, essentially comprising two sentences of overstated results.

This study provided the observational evidence confirming direct relationships between an increase in atmospheric DMS and the formation and growth of aerosol particles, and also an increase in the total mass concentration of nss-SO4– during Arctic phytoplankton blooms. Concurrent measurements of a suite of parameters (DMS, satellite-derived phytoplankton biomass, concentration and chemical composition of particles) supported the assertion that oceanic emission of DMS significantly affects the properties of sub-micrometer particles in the Arctic atmosphere.

I would think that the authors can only conclude:

During periods high biologicaly activity as illustrated by surrogates for plankton biomass, both enhancements of some aerosol-forming gaseous precursors (e.g. sulphate precursors such DMS and MSA) are seen simultaneous to a growing nucleation mode particles (new particle mode 3-10 nm) and by the time that these particles grow to sizes at which their chemical composition can be identified.

The chemical composition of marine aerosol cannot be resolved using the above instrumentation at sizes below 100 nm. In fact, I would suggest that given the mass mode in the size distribution is certainly larger than 500 nm and closer to 1,000 nm which means that the mass a particle has increased at least another three orders of magnitude when it has been chemically quantified.

---

## Author Comment (AC1) · 29 May 2017

We thank Referee 1 for providing numerous specific suggestions, which have considerably improved the readability of our revised manuscript. Our responses to this Referee's comments are presented below. The revised manuscript was uploaded in the form of a supplement.

Technical Comments

1. P2, lines 5 and 8. Check the references: We have added omitted citations to the revised list of references.

2. P2, line 12: We have changed "DMS eventually emitted" to "DMS is emitted".

[Figure]

3. P2, line 13: We have removed "rapidly" from line 13.

4. P2, line 14–16. Modify the description regarding Henry's constant and its association with particle nucleation: We agree with this referee that Henry's constants are macroscopic representations of particle formation, while particle nucleation is more a molecular-level process. Therefore, we made the following changes: "The MSA and $H_2SO_4$ formed from DMS tend to transform into new particles via multiple nucleation processes (i.e., binary, ternary, and ion-induced) or condense onto existing particles because of their low volatility nature".

5. P2, lines 23–25. Inappropriate use of "condensation": We have replaced "condensation of" with "the occurrence of large source of", and the text "because of the high level of production of condensable vapors and the relative lack of a condensation sink of pre-existing particles" has been added in lines 24–25.

6. P2: A research report by Leaitch et al. (2016) has been cited in line 27.

7. P3: Reports by Chang et al. (2011), Browse et al. (2012), Leaitch et al. (2013), Turnved et al. (2013), and Willis et al. (2016) have been cited in lines 3–4 (page 3), and the revised reference list.

8. P3. Provide more information about the use of DMA-CPC: In the revised manuscript (lines 19–23), we have stated that two discrete SMPS systems were used to measure the distribution of aerosol particle sizes in our study.

9. P4. Provide uncertainty for the $\delta34S$ value for sea salt: In the revised manuscript (P4, line 25), we have included a measure of uncertainty associated with the $\delta34S_{ss}$ value (21.0 +- 0.1‰ for sea salts. This uncertainty was estimated from direct measurements reported by Böttcher et al. (2007); this paper has been cited and added to the reference list as supporting evidence.

10. P5. Provide more description about the relation between atmospheric DMS mixing ratios and MSA concentration: We agree that some of the MSA, particularly that

measured in April, was probably formed elsewhere and subsequently transported to the measurement site. Local DMS production explained much of the variation in MSA in May, but did not account for all MSA variations observed in that month (see Fig. 1a and c). Part of the remaining variance in MSA in May can be explained by MSA that was introduced to the measurement site as a result of long-range transport, and the efficiency of photochemical oxidation of DMS. Therefore, we have added a short paragraph (lines 4–6, page 5) noting that transport of MSA to the site was a possibility.

11. P15: We have added the time series for solar irradiance to Figure 1b.

12. P5, line 16: We have replaced "probably" with "possibly".

13. P5, lines 18–20: We have cited O'Dowd et al. (2002) and Allan et al. (2015), and added a brief explanation to lines 18–19.

14. P6. Need an explicit explanation about changes in aerosol properties during the months of April and May: As this referee noted, our observations are not sufficient to prove that neutralization processes enhanced the growth of aerosol particles in the accumulation mode during the Arctic haze period. Therefore, to clarify the interpretation of our measurements we have revised the paper as follows (see lines 3–6, page 6): "The transition of aerosol microphysical properties from a distribution dominated by an accumulation mode (Arctic haze period) to a distribution dominated by nucleation and Aitken mode atmospheric particles (phytoplankton bloom period) was probably driven by the combination of three factors, including changes in air mass transport, incoming solar radiation and condensation sink processes (Tunved et al., 2004 and 2013)".

15. P6, line 8. Inappropriate reference: we have removed the Sharma et al. (2012) citation from the text and the reference list.

16. P6, lines 20–22: We have added a brief overview of recent field observations reported in Willis et al. 2016, which is conceptually similar to our work.

17. P6, line 20: We have removed "(promoted by the values of Henry's law constants

for H2SO4 and MSA)" from the revised text.

18. P6, line 25 and P18. We have moved Figure S3 to the main text.

19. P7, line 6. Be consistent with S isotope ratios of anthropogenic SO42– ($\delta$34Santh): We have replaced "(0–8‰ )" with "(5 $\pm$ 1‰ )".

20. P8. Need to acknowledge potential contribution of anthropogenic SO42– to NPF events during the phytoplankton bloom period: The contribution of anthropogenic SO42– to fine SO42– particles (< 2.5 $\mu$m in diameter) was still significant in May (30–60%). Unfortunately, we could not accurately estimate the relative contribution of biogenic versus anthropogenic SO42– to the formation and growth of aerosol particles. However, recent field observations indicate that a considerable amount of SO42– in aerosol particles having a diameter < 0.49 $\mu$m is biogenic (> 63%), based on size-segregated sulfur isotope analysis in the Arctic atmosphere. Therefore, we have added a short paragraph indicating the limitations of our study (lines 3–10, page 8), and cited Ghahremaninezhad et al. (2016) in support of our argument.

21. P8, line 28: We have removed the sentence including "confirming".

Please also note the supplement to this comment:
http://www.atmos-chem-phys-discuss.net/acp-2016-1161/acp-2016-1161-AC1-supplement.pdf

**Supplement:**

[revised manuscript text omitted]

---

## Author Comment (AC3) · 29 May 2017

[revised manuscript text omitted]

 The occurrence of a large source of low-volatility vapors (generally involving $H_2SO_4$) in the absence of coagulation with larger particles results in the constant formation of new particles in the atmosphere , because of the high level of production of condensable vapors and the relative lack of a condensation sink of pre-existing particles; these new particles subsequently grow into Aitken particles (> 50–100 nm in diameter), which probably influence cloud formation, and thereby
30  radiation (Boy et al., 2005; Pierce et al., 2014; Leaitch et al., 2016). However, it is difficult to establish the quantitative relationship between oceanic DMS emission and the formation and growth of aerosol particles in the marine boundary layer. A small number of recent studies have reported that atmospheric DMS mixing ratios are related to the ocean phytoplankton biomass (Preunkert et al., 2008; Park et al., 2013), and that an increase in nss-$SO_4^{2-}$ corresponds to a proportional increase in
* * *
**메모 포함[박3]:** (response to referee #1)
**Q1:** We have added omitted citations to the revised list of references.

**메모 포함[박4]:** (response to referee #2)
**Q3.** We have changed "*direct association*" to "*close linkage*".

**메모 포함[박5]:** (response to referee #2)
**Q13. Is there non-biogenic DMS?** More than 90% of DMS is derived from marine ecosystem. Other DMS sources are negligible.

**메모 포함[박6]:** (response to referee #2)
**Q13.** We have added this reference.

**메모 포함[박7]:** (response to referee #1)
**Q2.** As suggested, we have changed "DMS eventually emitted" to "DMS is emitted"

**메모 포함[박8]:** (response to referee #1)
**Q3.** As suggested, we have removed "rapidly".

**메모 포함[박9]:** (response to referee #1)
**Q4. Modify the description regarding Henry's constant and its association with particle nucleation:** Henry's constants are macroscopic representations of particle formation, while particle nucleation is more a molecular-level process. Therefore, we have revised the sentence.

**메모 포함[박10]:** (response to referee #2)
**Q4. The DMS-climate feedback may not as significant as other feedback processes:** We believe that the DMS-climate feedback mechanism is yet to be tested in the Arctic environment, where most rapid warming has occurred. In the revised manuscript we have more clearly elaborated the value of our study testing the DMS-climate feedback

**메모 포함[박11]:** (response to referee #2)
**Q3.** We have change "*a direct association*" to "*an association*".

**메모 포함[박12]:** (response to referee #1)
**Q5. Inappropriate use of "*condensation*":** We have replaced "*condensation of*" to "*the occurrence of large source of*".

**메모 포함[박13]:** (response to referee #1 and 2)
**Q5.** We have added this sentence.

**메모 포함[박14]:** (response to referee #1)
**Q6.** We have added this reference.

the MSA concentration in regions of high phytoplankton productivity (Becagli et al., 2012, 2013 and 2016; Zhang et al., 2015), where DMS emissions are also high. However, a mechanistic understanding of the major physical and chemical processes that are involved in the formation of DMS-derived $SO_4^{2-}$ particles and their growth into larger particles remains elusive. Explicitly, DMS emissions may exert greater impacts on aerosol formation in regions where the concentration of background aerosol particles is low, but DMS-producing phytoplankton are abundant. The Arctic atmosphere is an excellent example of an environment that meets these two criteria (e.g., Chang et al., 2011; Browse et al., 2012; Leaitch et al., 2013; Tunved et al., 2013; Willis et al., 2016).

The aims of the present study were to investigate  the possible association between DMS emissions and the formation of aerosol particles, and to assess the contribution of DMS to total $SO_4^{2-}$ aerosol budget. To this end we analyzed datasets of atmospheric DMS mixing ratio, aerosol particle size distributions, and aerosol chemical composition measured at Ny-Ålesund (Svalbard; 78.5° N, 11.8° E) in April and May 2015. To address the second aim we analyzed the MSA concentration (formed exclusively from the photo-oxidation of DMS) and the stable S isotope composition of aerosol particles.

**2 Experimental Methods**

The atmospheric DMS mixing ratio was measured at 1–2 h intervals on the Zeppelin observatory, which is located at an elevation of 474 m above sea level (m.a.s.l) and 2 km south and southwest of Ny-Ålesund. The measurement period (April – May) approximately covered the pre- to post-phytoplankton bloom periods. The analytical system includes a component for DMS trapping and elution and a gas chromatography (GC) equipped with a pulsed flame photometric detector (PFPD) enabling DMS quantification. The detection limit of the analytical DMS system was reported to be 1.5 pptv in an air volume of approximately 6 L. Details for the DMS analytical system and the DMS measurement protocol are described in Jang et al (2016).

The distribution of aerosol particle sizes was measured at the Gruvebadet observatory, which is approximately 1 km southwest of Ny-Ålesund and approximately 60 m.a.s.l.  Two discrete systems of scanning mobility particle sizer (SMPS) systems, each of which included a differential mobility analyzer (DMA) and a condensation particle counter (CPC), continuously measured the distribution of small particles in in differential mobility equivalent diameter ranges 3–60 nm (combination of TSI 3085 and TSI 3776) and 10–500 nm (TSI 3034), and an aerodynamic particle sizer (APS) analyzed larger particles in the range 0.5–20 µm in diameter (Park et al., 2014; Lupi et al., 2016).

A high volume air sampler equipped with a $PM_{2.5}$ impactor (collecting particles < 2.5 µm in aerodynamic equivalent diameter) was used for collection of aerosol samples. The sampler was mounted on the roof of the Gruvebadet observatory, and sampled

메모 포함[박15]: (response to referee #1)
**Q7.** We have added these references.

메모 포함[박16]: (response to referee #2)
General concern: we have changed "*direct association*" to "*possible association*".

메모 포함[박17]: (response to referee #1 and #2)
**Q8. Provide more information about the use of DMA-CPC:** Two discrete SMPS systems were used to measure the distribution of aerosol particle sizes. We have added more detailed information and references.

particles every 3 days between 9 April and 20 May 2015, and later measured concentrations of major ions and the stable S isotope composition on a quartz filter.

For measurement of major ions ($Na^+$, $K^+$, $Mg^{2+}$, $SO_4^{2-}$, $Cl^-$, and MSA), a 47-mm (diameter) disk filter was punched out from a $PM_{2.5}$ aerosol quartz filter. All major ions collected on the disk filter were extracted in 50 mL Milli-Q water and analyzed by a Dionex ion chromatography system (Thermo Fisher Scientific Inc., USA). The concentrations of major anions were determined using a Dionex model ICS-2000 with an IonPac AS 15 column and the concentrations of major cations were determined using a Dionex model ICS-2100 with an IonPac CS 12A column. Three times the standard deviations of blank measurements were used as detection limits (0.01 to 0.26 ng mL$^{-1}$) (Kang et al., 2015).

For measurement of the stable S isotope ratio ($^{34}S/^{32}S$), all S compounds on half of a $PM_{2.5}$ quartz filter were extracted in 50 mL Milli-Q water. The filtrate was treated with 50−100 µL of 1M HCl to adjust the solution to pH = 3−4. Then, 100 µL of 1M $BaCl_2$ was added to cause all S as $SO_4^{2-}$ to precipitate as $BaSO_4$. After a 24-h precipitation period at room temperature, the $BaSO_4$ precipitate was recovered by filtration on a membrane filter and finally dried for 24 h. Each membrane filter containing $BaSO_4$ was packed into a tin cup and analyzed by isotope ratio mass spectrometer (IsoPrime100, IsoPrime Ltd, UK) coupled to an elemental analyzer (Vario EL, Elementar Co, German). The resulting S isotope ratio of a sample ($\delta^{34}S$) was expressed as parts per thousand (‰) relative to the $^{34}S/^{32}S$ ratio in a standard (Vienna Cañon Diablo Troilite) (Krouse and Grinenko, 1991).

$$\delta^{34}S \ (‰) = \{(^{34}S/^{32}S)_{sample} / (^{34}S/^{32}S)_{standard} - 1\} \ X \ 1000 \tag{1}$$

Information about the S isotope ratio of aerosol particles and the concentrations of major ions enabled the estimation of the relative contributions of biogenic DMS ($f_{bio}$), anthropogenic $SO_X$ ($f_{anth}$), and sea-salt $SO_4^{2-}$ ($f_{ss}$) to the total aerosol $SO_4^{2-}$ concentration. The concentration of ss-$SO_4^{2-}$ was first calculated by multiplying the $Na^+$ concentration (as a sea spray marker) by 0.252 (the seawater ratio of $SO_4^{2-}/Na^+$) (Keene et al., 1986). The nss-$SO_4^{2-}$ fraction of the total $SO_4^{2-}$ was then calculated by subtracting the fraction of ss-$SO_4^{2-}$ from the total $SO_4^{2-}$. Finally, the fraction of biogenic $SO_4^{2-}$ was estimated by solving the following equations:

$$\delta^{34}S_{measured} = \delta^{34}S_{bio} \ f_{bio} + \delta^{34}S_{anth} \ f_{anth} + \delta^{34}S_{ss} \ f_{ss} \tag{2}$$

$$f_{bio} + f_{anth} + f_{ss} = 1 \tag{3}$$

$$f_{ss} = 0.252 \ Na^+/\text{total} \ SO_4^{2-} \tag{4}$$

In solving Eq. (2)–(4), the published S isotope end-member values of DMS-derived $SO_4^{2-}$ ($\delta^{34}S_{bio}$ = 18 ± 2‰), anthropogenic $SO_4^{2-}$ ($\delta^{34}S_{anth}$ = 5 ± 1‰), and sea-salt $SO_4^{2-}$ ($\delta^{34}S_{ss}$ = 21.0 ± 0.1‰) were used (McArdle and Liss, 1995; Norman et al., 1999; Böttcher et al., 2007; Lin et al., 2012).

메모 포함[박18]: (response to referee #1)
Q9. we have included a measure of uncertainty associated with the $\delta^{34}S_{ss}$ value (21.0 ± 0.1‰) for sea salts.

메모 포함[박19]: (response to referee #1)
Q9. We have added this reference.

**3 Results and Discussion**

**3.1 Atmospheric DMS and aerosol particles**

The atmospheric DMS mixing ratio measured at Zeppelin observatory changed abruptly (by several orders of magnitude) within a few days of measurement and occasionally reached a level of 400 pptv, particularly during phytoplankton bloom events (Fig. 1a). The monthly mean DMS mixing ratio for May (47 ± 91 pptv) was  more than double than the April mean (18 ± 18 pptv). The 3-day integrated concentrations of MSA were broadly consistent with the concentrations of DMS; the lowest concentration (< 50 ng m$^{-3}$) occurred in April and the highest value (approximately 200 ng m$^{-3}$) occurred in May (Fig. 1a). The strong positive correlation between the MSA concentrations and the corresponding DMS mixing ratios ($r$ = 0.84, $n$ = 14, P < 0.05; Fig.  1c) supports the assumption that the photochemical oxidation of biogenic DMS was the  major source of MSA in our study area. Variations in DMS explained approximately 70% of the observed variance in the MSA concentration; the remaining variance was probably associated with variations in MSA that formed elsewhere, and was subsequently advected to the measurement site, and also with variations in the efficiency of photochemical oxidation of DMS. The concentration of aerosol particles in the 3–10 nm diameter range (a nucleation mode), which is an indicator of recent nucleation, occasionally exceeded 3000 cm$^{-3}$. These small particles formed more frequently in May than in April (blue line in Fig. 1a).  The observed increase in nucleation mode particles coincided with high atmospheric DMS mixing ratio and MSA concentration. Therefore, the 3-day mean DMS mixing ratios and the MSA concentrations were both significantly correlated with the 3-day mean concentration of nucleation mode particles ($r$ = 0.66, $n$ = 14, P < 0.05, Fig.  1d; and $r$ = 0.71, $n$ = 14, P < 0.05, Fig.  1e, respectively) Approximately 45% of the variability in the 3-day mean concentrations of nucleation mode particles can be explained by overall variations in the concentrations of DMS and MSA; some of the remaining variability will be associated with variations in the intensity of solar radiation, which influences the efficiency of photochemical oxidation of DMS. In fact, high atmospheric DMS mixing ratios found in mid-April (77.1 ± 51.5 pptv; 14–17 April) was not followed by the formation of nucleation mode particles (42.6 ± 49.5 cm$^{-3}$) and MSA (21.4 ng m$^{-3}$),  possibly due to the low intensity of solar irradiation (80.4 ± 81.9 W m$^{-2}$) (Fig. 1a  , c and d). We cannot completely rule out the possibility that sources other than DMS  contributed to the formation of nucleation mode particles (Fig. 1a,  d and e). The emission of iodine is an alternative explanation for the particle nucleation event in the Arctic atmosphere during our study period (O'Dowd et al., 2002; Allan et al., 2015). Recent field observations in an iodine-rich coastal environment have shown that species containing iodine contribute to the formation of new aerosol particles via direct molecular-scale observations of nucleation in an iodine-rich coastal environment (Sipilä et al., 2016). As all chemical species (including H$_2$SO$_4$, iodine species, and organic vapors) that are directly involved in the nucleation process were not measured during the observational periods of the present study, we are unable to pinpoint the major contributor; however, these strong correlations indicate that the small aerosol particles that were formed newly were probably derived from recently released biogenic DMS.

메모 포함[박20]: (response to referee #2)
Q6. As suggested, we have changed "*3-fold higher*" to "*more than double*".

메모 포함[박21]: (response to referee #1)
Q10. Provide more description about the relation between atmospheric DMS mixing ratios and MSA concentration: Local DMS production explained much of the variation in MSA, but did not account for all MSA variations. Therefore, we have added a short paragraph noting that long range transport of MSA to the site was a possibility.

메모 포함[박22]: (response to referee #2)
Q7. Our description of Figure 1 was misleading. In the revised manuscript we have newly added this sentence.

메모 포함[박23]: (response to referee #2)
Q7. Our description of Figure 1 was misleading. In the revised manuscript we have removed this sentence.

메모 포함[박24]: (response to referee #1)
Q10. We have added "*and MSA (21.4 ng m$^{-3}$)*".

메모 포함[박25]: (response to referee #1)
Q12. We have replaced "*probably*" to "*possibly*".

메모 포함[박26]: (response to referee #1)
Q13. We have cited O'Dowd et al. (2002) and Allan et al. (2015), and added a brief explanation.

메모 포함[박27]: (response to referee #2)
Q8. Is DMS only responsible for the formation of small particles? Figure 1a shows that approximately 45% of the variance in small particle formation was explained by DMS. Iodine may have contributed to explaining the remaining variance. We have added statement that iodine could be an important contributor.

메모 포함[박28]: (response to referee #2)
Q9. This study did not provide direct evidence: We agree that molecular-scale measurements of chemical species actually involved in nucleation processes are required to provide direct evidence for the DMS-derived aerosol formation events. Therefore, we have added a short paragraph indicating the limitation of our study in this regard.

**3.2 Aerosol particles formed during periods of Arctic haze (April) and phytoplankton blooms (May)**

Arctic haze, formed originally from emissions of pollution in North Europe, Siberia and North America, and its transport to the Arctic environment, has been reported to influence the aerosol characteristics of the Arctic atmosphere during early spring (April and earlier periods). The Arctic haze is a mixture of $SO_4^{2-}$ and particulate organic matter, plus minor contributions of ammonium, nitrate, dust, black carbon, and heavy metals (Quinn et al., 2007). The concentrations of nss-$SO_4^{2-}$ during this period reached 2000 ng m$^{-3}$, and the mean level in April was 2-fold greater than that in May (Fig. 2a). However, information about the concentrations of nss-$SO_4^{2-}$ only did not enable differentiation of the strengths of two major sources (anthropogenic $SO_X$ vs. biogenic DMS). Additional measurements of particle concentrations enabled quantification of the contributions of anthropogenic $SO_X$ and biogenic DMS to the total nss-$SO_4^{2-}$.

Anthropogenic $SO_X$ from the continents that reaches Ny-Ålesund will have undergone alterations during long-range transport. For example, hydrophilic $H_2SO_4$ particles derived from anthropogenic $SO_X$ tend to grow into particles larger than 100 nm (accumulation mode), especially following partial or total neutralization by $NH_3$, which in turn yields hygroscopic compounds such as $NH_4HSO_4$ and $(NH_4)_2SO_4$. In contrast, the photochemical oxidation of DMS (yielding biogenic $SO_4^{2-}$) occurs locally and the resulting biogenic particles more tend to develop into small particles (nucleation mode, 3–10 nm; and Aitken mode, 10–100 nm). These established explanations are generally consistent with our observations. The transition of aerosol microphysical properties from a distribution dominated by an accumulation mode (Arctic haze period) to a distribution dominated by nucleation and Aitken mode atmospheric particles (phytoplankton bloom period) was probably driven by the combination of three factors, including changes in air mass transport, incoming solar radiation and condensation sink processes (Tunved et al., 2004 and 2013). Specifically, the large accumulation mode particles outnumbered the small nucleation and Aitken mode particles during early spring (April) but the concentration of those large particles decreased rapidly from April to May, with particles smaller 100 nm becoming dominant in May (Fig. 3 and S1). The high concentration of small particles (< 100 nm) during phytoplankton bloom period (May) constitutes compelling evidence for new particle formation derived from local DMS emission (Sharma et al., 2012; Tunved et al., 2013). A similar sharp transition (large-to-small particles) in the dominant particle type was also identified in previous observations at the same site (Engvall et al., 2008).

Our data on the particle size distributions showed that particles > 100 nm were more abundant in April, whereas small particles (< 100 nm) were more abundant in May (Fig. 3 and S1). As a result, the total surface area of aerosol particles in April was 2-fold greater than that observed in May, whereas the concentration of particles in April was 3-fold less than that in May (Fig. 3). As the condensation sink is proportional to the surface area of aerosol particles, it will decrease with decreasing intensity of Arctic haze. The concentrations of nss-$SO_4^{2-}$ measured in April did not correlate with the levels of biogenic MSA (P > 0.05; blue circles in Fig. 2b). On the contrary, a strong correlation ($r = 0.75$, $n = 7$, P < 0.05; red circles in Fig. 2b) between these two parameters was found in May. The greater DMS contribution to the formation of nss-$SO_4^{2-}$ in May than in April is broadly consistent with the 2-fold greater chlorophyll concentration observed in May compared to April (Fig. S2). These observations support our hypothesis that the formation of new particles resulting from the photo-oxidation of biogenic DMS, followed by a

**메모 포함[박29]:** (response to referee #1)
**Q14. Need an explicit explanation about changes in aerosol properties during the months of April and May:** As this referee noted, our observations are not sufficient to prove that neutralization processes enhanced the growth of aerosol particles in the accumulation mode during the Arctic haze period. Therefore, to clarify the interpretation of our measurements.

**메모 포함[박30]:** (response to referee #1)
**Q14 and 15.** We have removed this sentence and references.

**메모 포함[박31]:** (response to referee #2)
**Q11. What role had super-micron particles in surface area or as a condensation sink?** Because super-micron particles have the greater surface area than sub-micron particles, and the condensation sink is proportional to the surface area of aerosol particles, the increase in super-micron particles would be expected to increase the condensation sink and depress the nucleation rate. We have added a short paragraph explaining the link between increase in super-micron particles and decrease in the nucleation rate

**메모 포함[박32]:** (response to referee #2)
**Q12.** We have removed "*hypothesis*"

gas-to-particle conversion (promoted by the values of Henry's law constants for H₂SO₄ and MSA), is an important source of Aitken mode (10–100 nm) particles in the Arctic atmosphere. Recent field observation also provided the evidence that the growth of nucleation mode particles in the summertime Arctic atmosphere can be mediated by the presence of secondary marine organic aerosol including MSA (Willis et al., 2016).

**3.3   $SO_4^{2-}$ aerosol particles formed from biogenic DMS**

The use of an asymptotic value in a plot of nss-$SO_4^{2-}$/MSA ratio versus MSA concentration is a convenient method for estimating the fraction of biogenic $SO_4^{2-}$ aerosols. As the MSA concentration in an aerosol sample increases, the contribution of biogenic $SO_4^{2-}$ to the total nss-$SO_4^{2-}$ will also increase while the contributions of other sources will decrease. In this case, the nss-$SO_4^{2-}$/MSA ratio tends to approach an asymptotic value as the MSA concentration increases. Therefore, this asymptotic value adequately represents the nss-$SO_4^{2-}$/MSA ratio derived exclusively from DMS (Udisti et al., 2012, 2016). The biogenic $SO_4^{2-}$/MSA ratio has been reported to vary considerably in space and time (Gondwe et al., 2004), because the ratio is sensitive to temperature, and to a lesser extent photochemical species or reactions with halogen radicals (Bates et al., 1992). When this method was applied to data for aerosol samples ($PM_{10}$) collected in 2014 at the Gruvebadet observatory, in a vicinity of our DMS measurement site, the biogenic $SO_4^{2-}$/MSA ratio was estimated to be 3.0 (Udisti et al., 2016) ( Fig. 4). In other polar locations, a ratio of 2.6 was reported, including for Alert station (82.5° N, 62.3° W; 210 m.a.s.l.) (Norman et al., 1999) and Concordia station (75.1° S, 123.3° E; 3233 m.a.s.l) (Udisti et al., 2012). We estimated the amount of biogenic $SO_4^{2-}$ by multiplying the biogenic $SO_4^{2-}$/MSA ratio (3.0) by the MSA concentration in each aerosol sample. The fraction of anthropogenic $SO_4^{2-}$ was estimated by subtracting the combined ss-$SO_4^{2-}$ plus biogenic $SO_4^{2-}$ concentration from the total $SO_4^{2-}$ concentration.

Another method for estimating biogenic $SO_4^{2-}$ is to use S isotope ratios ($\delta^{34}S$) of $SO_4^{2-}$ aerosols, because the $\delta^{34}S$ values of biogenic DMS (18 ± 2‰) are greater than those of anthropogenic $SO_4^{2-}$  (5 ± 1‰) but less than that of sea salt (21 ± 0.1‰) (e.g., Wadleigh, 2004; Lin et al., 2011, Oduro et al., 2012). A wide range in $\delta^{34}S$ (0–8‰) has been reported for anthropogenic $SO_2$ compared with values reported for other sources (Krouse and Grinenko, 1991). Surprisingly, Patris et al (2000) reported consistent regional-scale $\delta^{34}S$ values for anthropogenic $SO_2$. For example, in remote Arctic regions (including Ny-Ålesund and Alert) the S isotope ratios measured for $SO_4^{2-}$ aerosols during the Arctic haze period in a single year mostly fell within the narrow range of 5–6‰ (McArdle and Liss, 1995; Norman et al., 1999), probably because regional-scale mixing processes averaged the signals (Partis et al., 2000). During the study period the $\delta^{34}S$ values measured at Ny-Ålesund ranged from 4.6 to 10.3. The $\delta^{34}S$ values were higher in May (8.8–10.3‰) than in April (4.6–8.2‰), reflecting changes in S sources. The contributions of anthropogenic and biogenic $SO_4^{2-}$ to the total $SO_4^{2-}$ aerosols, estimated using two independent methods, are shown in  Fig 5 (and Fig.  S3). In April and May the contribution of ss-$SO_4^{2-}$ to total $SO_4^{2-}$ was small (< 3% of total aerosol particles < 2.5 μm in diameter). It was estimated that approximately 90% of the total $SO_4^{2-}$ was of anthropogenic origin in April, when the Arctic haze was most intense. This estimation is consistent with measurements of anthropogenic $SO_4^{2-}$ in $PM_{10}$ aerosols collected in April 2014 at the same site (Udisti et al., 2016). In May, following the period of Arctic

메모 포함[박33]: (response to referee #1)
Q17. We have removed "*(promoted by the values of Henry's law constants for H₂SO₄ and MSA)*" from the revised text.

메모 포함[박34]: (response to referee #1)
Q16. We have added a brief overview of recent field observations reported in Willis et al. 2016, which is conceptually similar to our work.

메모 포함[박35]: **The title of section 3.3 is inadequate:** We have changed "*Aerosol particles formed from biogenic DMS*" to "*SO₄²⁻ aerosol particles formed from biogenic DMS*"

메모 포함[박36]: (response to referee #1)
Q18. We have moved Figure S3 to the main text.

메모 포함[박37]: (response to referee #1)
Q19. **Be consistent with S isotope ratios of anthropogenic SO₄²⁻ (δ³⁴S_anth):** We have replaced "*(0–8‰)*" to *(5 ± 1‰)*".

haze, the contribution of DMS-derived $SO_4^{2-}$ sharply increased up to 70% of fine $SO_4^{2-}$ particles (< 2.5 µm in diameter), and this corresponded to the Arctic phytoplankton bloom (Fig. 4a and S4 5a and S3). The concentrations of DMS-derived $SO_4^{2-}$ estimated using the MSA-based approach were approximately 50% higher than the values based on the S-isotope method. The differences may be the result of uncertainties associated with uncertainties in assigned S isotope end-member values, and the

5  ratio of biogenic $SO_4^{2-}$ to MSA. Nonetheless, a robust correlation between these two estimates was found during April and May ($r = 0.74$, $n = 13$, P < 0.05; Fig. 4b 5b).

**3.4 Relationship between concentrations of DMS-derived $SO_4^{2-}$ and the concentrations of aerosol particles**

In both April and May, the concentrations of biogenic $SO_4^{2-}$ particles estimated using the MSA-based and S-isotope-based methods were significantly correlated with the concentration of small aerosol particles in nucleation (from 3–10 nm, $r = 0.71$,

10  $n = 14$, P < 0.05; Fig. 5a 6a) and Aitken modes (from 10–100 nm, $r = 0.89$, $n = 14$, P < 0.05; Fig. 5b 6b). However, when the Arctic haze prevailed in April, the concentration of large particles in accumulation mode (from 100 nm − 1 µm, P > 0.05; Fig. 5c 6c) was not significantly correlated with the concentration of biogenic $SO_4^{2-}$, but was strongly correlated with the concentration of anthropogenic $SO_4^{2-}$ ($r = 0.92$, $n = 7$, P < 0.05; the inset in Fig. 5c 6c). In contrast, when Arctic haze moderated in May and the abundance of phytoplankton began to increase, the concentration of biogenic $SO_4^{2-}$ was strongly correlated

15  with the concentration of accumulation mode particles ($r = 0.91$, $n = 7$, P < 0.05; Fig. 5c 6c). A strong correlation between biogenic $SO_4^{2-}$ particles and the surface areas of particles provided additional evidence that biogenic $SO_4^{2-}$ significantly contributed to small particle formation (Fig. S5 S4). Moreover, the formation and growth of sub-micrometer particles with diameters between 3 and 100 nm coincided with high concentrations of biogenic $SO_4^{2-}$ (Fig. 6 7 and S6 S5). It is noteworthy that the contribution of anthropogenic $SO_4^{2-}$ to fine $SO_4^{2-}$ particles (< 2.5 µm in diameter) was still considerable (30–60%)

20  during the bloom period (May) (Fig. 5a and S3). Therefore, in May anthropogenic $SO_4^{2-}$ may also have partly contributed to the formation and growth of small aerosol particles (< 100 nm). However, we could not accurately estimate the exact contributions of biogenic versus anthropogenic $SO_4^{2-}$ to the formation and growth of aerosol particles in the absence of information on the chemical composition of size-segregated aerosol particles. The chemical composition of size-segregated aerosol particles was recently measured in the Arctic atmosphere during summer months. More than 60% of the aerosol

25  particles havoing a diameter < 0.49 µm was found to be derived from biogenic $SO_4^{2-}$ (Ghahremaninezhad et al., 2016). According to a study based on an aerosol microphysics box model (Chang et al., 2011), the atmospheric DMS mixing ratios observed during phytoplankton bloom periods in our study were sufficiently high for the formation of ultrafine aerosol particles, when background particle concentrations are low (i.e. DMS mixing ratio > 100 pptv; condensation sink < 7.0 m$^{-2}$) (Fig. 1a and b).

30  These direct observations provide  evidence that biogenic DMS released from the Arctic Ocean contributed  to the formation of new aerosol particles, and their subsequent growth to larger climate-relevant particles (> 50–100 nm in diameter). In particular, new particles appeared to be formed more rapidly in May, and grew more efficiently to large climate-relevant particles, possibly because of the presence of a large source of condensable vapor (generally involving

메모 포함[박38]: (response to referee #1)
**Q20. Need to acknowledge potential contribution of anthropogenic $SO_4^{2-}$ to NPF events during the phytoplankton bloom period:** Unfortunately, we could not accurately estimate the relative contribution of biogenic versus anthropogenic $SO_4^{2-}$ to the formation and growth of aerosol particles. However, recent field observations indicate that a considerable amount of $SO_4^{2-}$ in aerosol particles having a diameter < 0.49 µm is biogenic (> 63%), based on size-segregated sulfur isotope analysis in the Arctic atmosphere. Therefore, we have added a short paragraph indicating the limitations of our study, and cited Ghahremaninezhad et al. (2016) in support of our argument.

메모 포함[박39]: (response to referee #2)
**Q10. Need more evidence (such as box model) to support strong statements regarding DMS-derived aerosol formation:** As supporting evidence that DMS can substantially contribute to the fine-mode particle formation we observed in the Arctic atmosphere, we have cited the modeling work of Chang et al. (2011), who showed that atmospheric DMS mixing ratios > 100 pptv are sufficient to account for the formation of ultrafine particles, particularly when background particle concentrations are low (condensation sink < 7.0 m$^{-2}$). To support the confidence of our statement, we have added a short paragraph describing the results of Chang et al. (2011) and have added this paper to the revised list of references.

메모 포함[박40]: (response to referee #2)
We have removed "*compelling*".

메모 포함[박41]: (response to referee #2)
We have removed "significantly".

[revised manuscript text omitted]

메모 포함[박66]: (response to referee #1)
**Q11.** As suggested, we have added the time series for solar irradiance to Figure 1b.

메모 포함[박67]: (response to referee #2)
**Q10.** We have added the calculated condensation sink.

[Figure]

**Figure 2:** (a) The concentrations of nss-SO$_4^{2-}$ (blue line), and MSA (red line) at the Gruvebadet observatory in April and May 2015. (b) The relationship between nss-SO$_4^{2-}$ and MSA measured in April (blue circles; Arctic haze period) and in May (red circles; phytoplankton bloom period). The red solid line shown in Fig. 2b indicates the best fit.

[Figure]

**Figure 3:** (a) Concentration, and (b) surface area of aerosol particles, including nucleation mode (3–10 nm), Aitken mode (10–100 nm), accumulation mode (100 nm–1 µm) and coarse mode (1–19 µm) particles in April and May 2015.

[Figure]

**Figure 4:** Relation between the nss-$SO_4^{2-}$/MSA ratio and the MSA concentration (PM$_{2.5}$; April and May 2015; grey circles). The dashed line indicates the bio-$SO_4^{2-}$/MSA ratio measured at Ny-Ålesund in 2014 (Udisti et al., 2016).

메모 포함[박68]: (response to referee #1)
**Q18.** We have moved Figure S3 to the main text.

[Figure]

**Figure 4 5:** Biogenic (bio) $SO_4^{2-}$ estimated using MSA and S isotope. (a) Biogenic $SO_4^{2-}$ as a percentage of the total aerosol $SO_4^{2-}$ burden. Black bars: biogenic $SO_4^{2-}$ estimated using MSA; grey bars: biogenic $SO_4^{2-}$ estimated using stable S isotope.
5    (b) The relationship between the concentrations of biogenic $SO_4^{2-}$ estimated using MSA, and the concentrations of biogenic $SO_4^{2-}$ estimated using stable S isotope. The black solid line represents the best fit.

[Figure]

**Figure 5 6:** Relationships between the concentration of biogenic (bio) $SO_4^{2-}$ and the concentrations of particles of various sizes in April and May. (a) Particles 3–10 nm, (b) particles 10–100 nm, and (c) particles 100 nm–1 μm. The black solid lines in Fig. 5a and 5b 6a and 6b indicate the best fit between biogenic $SO_4^{2-}$ and the particle concentration during April and May 2015. Blue and red circles in Fig. 5c 6c indicate data obtained in April (Arctic haze period) and May (phytoplankton bloom period), respectively. The red solid line in Fig. 5c 6c indicates the best fit between biogenic $SO_4^{2-}$ and particle concentration in May 2015, and the blue solid line in the inset of Fig. 5c 6c indicates the best fit between anthropogenic (anth) $SO_4^{2-}$ and particle concentration in April 2015.

[Figure]

**Figure 6 7:** Spectral plot of number size distribution (dN/dlogD$_p$) as a function of particle diameter (D$_p$, 10–500 nm) and year day during April and May 2015. The black line represents the concentration of biogenic SO$_4^{2-}$.

---

## Author Response (AR1)

**Response to Referee 1**

We thank Referee 1 for providing numerous specific suggestions, which have considerably improved the readability of our revised manuscript. Our responses to this Referee's comments are presented below.

**Technical Comments**

**1. P2, lines 5 and 8. Check the references:** We have added omitted citations to the revised list of references.

**2**. **P2, line 12:** We have changed "*DMS eventually emitted*" to "*DMS is emitted*".

**3**. **P2, line 13:** We have removed "*rapidly*" from line 13.

**4. P2, line 14–16. Modify the description regarding Henry's constant and its association with particle nucleation:** We agree with this referee that Henry's constants are macroscopic representations of particle formation, while particle nucleation is more a molecular-level process. Therefore, we made the following changes: "*The MSA and H$_2$SO$_4$ formed from DMS tend to transform into new particles via multiple nucleation processes (i.e., binary, ternary, and ion-induced) or condense onto existing particles because of their low volatility nature*".

**5. P2, lines 23–25. Inappropriate use of "*condensation*":** We have replaced "*condensation of*" with "*the occurrence of large source of*", and the text "*because of the high level of production of condensable vapors and the relative lack of a condensation sink of pre-existing particles*" has been added in lines 24–25.

**6. P2:** A research report by Leaitch et al. (2016) has been cited in line 27.

**7**. **P3:** Reports by Chang et al. (2011), Browse et al. (2012), Leaitch et al. (2013), Turnved et al. (2013), and Willis et al. (2016) have been cited in lines 3–4 (page 3), and the revised reference list.

**8. P3. Provide more information about the use of DMA-CPC:** In the revised manuscript (lines 19–23), we have stated that two discrete SMPS systems were used to measure the distribution of aerosol particle sizes in our study.

**9. P4. Provide uncertainty for the $\delta^{34}$S value for sea salt:** In the revised manuscript (P4, line 25), we have included a measure of uncertainty associated with the $\delta^{34}$S$_{ss}$ value ($21.0 \pm 0.1‰$) for sea salts. This uncertainty was estimated from direct measurements reported by Böttcher et al. (2007); this paper has been cited and added to the reference list as supporting evidence.

**10. P5. Provide more description about the relation between atmospheric DMS mixing ratios and MSA concentration:** We agree that some of the MSA, particularly that measured in April, was probably formed elsewhere and subsequently transported to the measurement site. Local DMS production explained much of the variation in MSA in May, but did not account for all MSA variations observed in that month (see Fig. 1a and c). Part of the remaining variance in MSA in May can be explained by MSA that was introduced to the measurement site as a result of long-range transport, and the efficiency of photochemical oxidation of DMS. Therefore, we have added a short paragraph (lines 4–6, page 5) noting that transport of MSA to the site was a possibility.

**11. P15:** We have added the time series for solar irradiance to Figure 1b.

**12. P5, line 16:** We have replaced "*probably*" with "*possibly*".

**13. P5, lines 18–20:** We have cited O'Dowd et al. (2002) and Allan et al. (2015), and added a brief explanation to lines 18–19.

**14. P6. Need an explicit explanation about changes in aerosol properties during the months of April and May:** As this referee noted, our observations are not sufficient to prove that neutralization processes enhanced the growth of aerosol particles in the accumulation mode during the Arctic haze period. Therefore, to clarify the interpretation of our measurements we have revised the paper as follows (see lines 3–6, page 6): "*The transition of aerosol microphysical properties from a distribution dominated by an accumulation mode (Arctic haze period) to a distribution dominated by nucleation and Aitken mode atmospheric particles (phytoplankton bloom period) was probably driven by the combination of three factors, including changes in air mass transport, incoming solar radiation and condensation sink processes (Tunved et al., 2004 and 2013)".*

**15. P6, line 8. Inappropriate reference:** we have removed the Sharma et al. (2012) citation from the text and the reference list.

**16. P6, lines 20–22:** We have added a brief overview of recent field observations reported in Willis et al. 2016, which is conceptually similar to our work.

**17. P6, line 20:** We have removed "*(promoted by the values of Henry's law constants for $H_2SO_4$ and MSA)*" from the revised text.

**18. P6, line 25 and P18.** We have moved Figure S3 to the main text.

**19. P7, line 6. Be consistent with S isotope ratios of anthropogenic $SO_4^{2-}$ ($\delta^{34}S_{anth}$):** We have replaced "*(0–8‰)*" with "*(5 ± 1‰)*".

**20. P8. Need to acknowledge potential contribution of anthropogenic $SO_4^{2-}$ to NPF events during the phytoplankton bloom period:** The contribution of anthropogenic $SO_4^{2-}$ to fine $SO_4^{2-}$ particles (< 2.5 µm in diameter) was still significant in May (30–60%). Unfortunately, we could not accurately estimate the relative contribution of biogenic versus anthropogenic $SO_4^{2-}$ to the formation and growth of aerosol particles. However, recent field observations indicate that a considerable amount of $SO_4^{2-}$ in aerosol particles having a diameter < 0.49 µm is biogenic (> 63%), based on size-segregated sulfur isotope analysis in the Arctic atmosphere. Therefore, we have added a short paragraph indicating the limitations of our study (lines 3–10, page 8), and cited Ghahremaninezhad et al. (2016) in support of our argument.

**21. P8, line 28:** We have removed the sentence including "*confirming*".

**Response to Referee 2**

We thank Referee 2 for providing insightful suggestions, which have considerably improved the readability of the revised manuscript. Our responses to this referee's one general and several comments are stated below.

**A general (major) concern**

**Overstatement of our results and conclusions:** A set of unique concurrent measurements (including atmospheric DMS and MSA concentrations, S isotope composition of aerosols, aerosol particle concentrations, and satellite-based biomass) that we made during Arctic phytoplankton bloom periods provided compelling evidence that there is a connection between DMS emissions and $SO_4^{2-}$ aerosol formation in the Arctic atmosphere. However, our measurements did not provide direct evidence supporting the connection between DMS emissions and the formation and growth of aerosol particles. We agree with Referee 2 that to directly confirm that DMS emissions lead to new particle formation, measurements of inorganic, organic, and halogen species are also needed. Therefore, we have amended the revised manuscript (Abstract, Results, and Conclusion) by changing the text reading "*the direct association of DMS emissions with the formation of aerosol particles*" to "*the significant association between DMS emissions and the formation of submicron $SO_4^{2-}$ aerosols*" (lines 19–20, page 1; lines 28–29, page 8) and "*possible association between DMS emissions and the formation of aerosol particles*" (lines 5–6, page 3). We have also noted that confirmation of this proposed link requires measurements of inorganic, organic, and halogen species (lines 22–25, page 5; lines 1–4, page 9).

**Specific Comments**

**1.** We have removed '*ocean*' from the title

**2. Abstract. Tone down the conclusion in our results reading "the direct association of DMS emissions with the formation and growth of aerosol particles":** As stated in our response to the general concern above, we have changed the above statement to "*the formation of submicron $SO_4^{2-}$ aerosols was significantly associated with an increase in the atmospheric DMS mixing ratio*" (lines 19–20, page 1).

**3. Introduction. A direct link between marine biota and climate change may not be accurate:** The connection between marine biota and climate change is far from simple, because multiple and complicating processes are interwoven, and consequently our assertion of a "direct link" may not appropriate. Therefore, we have changed "*direct association*" to "*close linkage*" (lines 9–10, page 2) and "*….a direct association….*" to "*….an association….*" (lines 19–20, page 2).

**4. Introduction. The DMS-climate feedback may not as significant as other feedback processes, for example the iodine-feedback:** We believe that the DMS-climate feedback mechanism is yet to be tested in the Arctic environment, where most rapid warming has occurred. The critical testing has not been possible in the Arctic, mainly because of the absence of atmospheric DMS data. The absence of data necessary to perform this critical testing greatly enhances the utility of our present

study. In the revised manuscript we have more clearly elaborated the value of our study testing the DMS-climate feedback (lines 17–19, page 2).

We agreed with Referee 2 that chemical species other than DMS may be equally or even more important than DMS in climate feedback. We have addressed this issue in response 8 below.

5. **Page 3. Provide more information about the online aerosol size distribution measurements:** We have addressed this issue in our response (8) to comments of Referee 1 (see lines 19–23, page 3).

6. **Page 4, line 31:** We have replaced "*3-fold higher*" with "*more than double*".

7. **Page 5, lines 9–10. The greater formation of nucleation mode particles in May than in April is not a direct cause of the strong correlation between DMS/MSA and nucleation mode particles:** Our description of Figure 1 was misleading. In the revised manuscript we have changed "*the observed nucleation events also concurrently occurred with high atmospheric DMS mixing ratios*" to "*The observed increase in nucleation mode particles coincided with high atmospheric DMS mixing ratio and MSA concentration*".

8. **Page 5. Is DMS only responsible for the formation of small particles?** Figure 1a shows that approximately 45% of the variance in small particle formation was explained by DMS. This does not mean that DMS was the only contributor to small particle formation. Iodine may have contributed to explaining the remaining variance. We explicitly stated in lines 18–22 that iodine could be an important contributor, as demonstrated in an iodine-rich coastal environment.

9. **Page 5. This study did not provide direct evidence that DMS-derived sulfate was responsible for the formation of nucleation mode particles:** We agree that molecular-scale measurements of chemical species actually involved in nucleation processes are required to provide direct evidence for the DMS-derived aerosol formation events. Therefore, we have added a short paragraph indicating the limitation of our study in this regard (lines 22–25, page 5; lines 1–4, page 9).

10. **Page 6. Need more evidence (such as box model) to support strong statements regarding DMS-derived aerosol formation:** As supporting evidence that DMS can substantially contribute to the fine-mode particle formation we observed in the Arctic atmosphere, we have cited the modeling work of Chang et al. (2011), who showed that atmospheric DMS mixing ratios > 100 pptv are sufficient to account for the formation of ultrafine particles, particularly when background particle concentrations are low (condensation sink < 7.0 m$^{-2}$). To support the confidence of our statement, we have added a short paragraph (lines 10–13, page 8) describing the results of Chang et al. (2011) and have added this paper to the revised list of references.

11. **Page 6. What role had super-micron particles in surface area or as a condensation sink?** Because super-micron particles have the greater surface area than sub-micron particles, and the condensation sink is proportional to the surface area of aerosol particles, the increase in super-micron particles would be expected to increase the condensation sink and depress the nucleation rate. We have added a short

paragraph explaining the link between increase in super-micron particles and decrease in the nucleation rate (line 25, page 2 and lines 14–15, page 6).

12. **P6, line 19:** We have removed "*hypothesis*" from line 19, page 6.

13. **Is there non-biogenic DMS?** More than 90% of DMS is derived from marine ecosystem. Other DMS sources are negligible. Reference to the publications of Kettle and Andreae (2000) and Stefels et al. (2007) have been added to support the assertion that almost all DMS was of marine origin (lines 10–11, page 2).

14. **The title of section 3.3 "Aerosol formation from biogenic DMS" is inadequate:** Because the data presented in the present study does not provide evidence supporting a direct link between aerosol particle formation and DMS-derived sulfate aerosol, we have changed "*Aerosol particles formed from biogenic DMS*" to "*$SO_4^{2-}$ aerosol particles formed from biogenic DMS*" (line 23, page 6).

15. **Is there closure between the SMPS/DMPS/APS derived mass distributions, or integrated mass distributions up to 2.5 microns and sulphate/seasalt?** This issue was beyond the scope of the present study, and our data did not enable it to be addressed because the total mass of the aerosol samples and the concentration of organic carbon collected on the PM2.5 filters were not analyzed.

16. **How much of the DMS is available for nucleation/condensation of sulphuric acid to the aerosol phase and over what timescale?** This question was also beyond the scope of the present study. The oxidation rate of atmospheric DMS in our study area could not be resolved because we did not concurrently measure key parameters (e.g., halogen species) affecting the conversion efficiency of DMS to $SO_2$. Therefore, we can only say that an atmospheric DMS mixing ratio >100 ppt might be sufficient for the formation of ultrafine aerosol particles when background particle concentrations are low in the remote Arctic marine boundary layer. We have addressed this issue in our response 10.

17. **Our results were overly stated in Conclusion. We have already addressed this issue in our response to a Referee 2's general concern.** We have clearly stated in the revised Conclusion (line 28, page 8 – line 4, page 9) that our measurements (atmospheric DMS and MSA concentrations, S isotope composition of aerosols, aerosol particle concentrations, and satellite-based biomass) were not direct evidence supporting a connection between DMS emissions and $SO_4^{2-}$ aerosol formation in the Arctic atmosphere.

**Observational evidence for the formation of  DMS-derived aerosols during Arctic phytoplankton blooms**

Ki-Tae Park[1,†], Sehyun Jang[2,†], Kitack Lee[2,*], Young Jun Yoon[1,*], Min-Seob Kim[3], Kihong Park[4], Hee-Joo Cho[4], Jung-Ho Kang[1], Roberto Udisti[5], Bang-Yong Lee[1], Kyung-Hoon Shin[6]

[1]Korea Polar Research Institute, Incheon, 21990, Korea
[2]Division of Environmental Science and Engineering, Pohang University of Science and Technology, Pohang, 37673, Korea
[3]Environment Measurement & Analysis Center, National Institute of Environmental Research, Incheon, 22689, Korea
[4]School of Environmental Science and Engineering, Gwangju Institute of Science and Technology, Gwangju, 61005, Korea
[5]Department of Chemistry, University of Florence, Florence, 50016, Italy
[6]Department of Marine Sciences and Convergent Technology, Hanyang University, Ansan 1588, Korea

*Correspondence to*: Kitack Lee (ktl@postech.ac.kr), Young Jun Yoon (yjyoon@kopri.re.kr)

†These authors (Ki-Tae Park and Sehyun Jang) contributed equally

**Abstract.** The connection between marine biogenic dimethyl sulfide (DMS) and the formation of aerosol particles in the Arctic atmosphere was evaluated by analyzing atmospheric DMS mixing ratios, aerosol particle size distributions and aerosol chemical composition data that were concurrently collected at Ny-Ålesund, Svalbard (78.5° N, 11.8° E) during April and May 2015. Measurements of aerosol sulfur (S) compounds showed distinct patterns during periods of Arctic haze (April) and phytoplankton blooms (May). Specifically, during the phytoplankton bloom period the contribution of DMS-derived $SO_4^{2-}$ to the total aerosol $SO_4^{2-}$ increased by 7-fold compared with that during the proceeding Arctic haze period, and accounted for up to 70% of fine $SO_4^{2-}$ particles (< 2.5 µm in diameter).  The results also showed that the formation of submicron $SO_4^{2-}$ aerosols was significantly associated with an increase in the atmospheric DMS mixing ratio. More importantly, two independent estimates of the formation of DMS-derived $SO_4^{2-}$ aerosols, calculated using the stable S isotope ratio and the non sea salt $SO_4^{2-}$/methanesulfonic acid ratio, respectively, were in close agreement, providing compelling evidence that the contribution of biogenic DMS to the formation of aerosol particles was substantial during the Arctic phytoplankton bloom period.

메모 포함[박1]: (response to referee #2)
Q1. We have removed '*ocean*' from the title.

메모 포함[박2]: (response to referee #2)
Q2. Overstatement of our results and conclusions: We agree that our measurements did not provide direct evidence supporting a connection between DMS emissions and the formation and growth of aerosol particles. Therefore, we have revised the sentence.

**1 Introduction**

[revised manuscript text omitted]

메모 포함[박3]: (response to referee #1)
Q1: We have added omitted citations to the revised list of references.

메모 포함[박4]: (response to referee #2)
Q3. We have changed "*direct association*" to "*close linkage*".

메모 포함[박5]: (response to referee #2)
Q13. Is there non-biogenic DMS? More than 90% of DMS is derived from marine ecosystem. Other DMS sources are negligible.

메모 포함[박6]: (response to referee #2)
Q13. We have added this reference.

메모 포함[박7]: (response to referee #1)
Q2. As suggested, we have changed "DMS eventually emitted" to "DMS is emitted"

메모 포함[박8]: (response to referee #1)
Q3. As suggested, we have removed "rapidly".

메모 포함[박9]: (response to referee #1)
Q4. Modify the description regarding Henry's constant and its association with particle nucleation: Henry's constants are macroscopic representations of particle formation, while particle nucleation is more a molecular-level process. Therefore, we have revised the sentence.

메모 포함[박10]: (response to referee #2)
Q4. The DMS-climate feedback may not as significant as other feedback processes: We believe that the DMS-climate feedback mechanism is yet to be tested in the Arctic environment, where most rapid warming has occurred. In the revised manuscript we have more clearly elaborated the value of our study testing the DMS-climate feedback

메모 포함[박11]: (response to referee #2)
Q3. We have change "a direct association" to "an association".

메모 포함[박12]: (response to referee #1)
Q5. Inappropriate use of "*condensation*": We have replaced "*condensation of*" to "*the occurrence of large source of*".

메모 포함[박13]: (response to referee #1 and 2)
Q5. We have added this sentence.

메모 포함[박14]: (response to referee #1)
Q6. We have added this reference.

메모 포함[박15]: (response to referee #1)
Q7. We have added these references.

메모 포함[박16]: (response to referee #2)
General concern: we have changed "*direct association*" to "*possible association*".

with a pulsed flame photometric detector (PFPD) enabling DMS quantification. The detection limit of the analytical DMS system was reported to be 1.5 pptv in an air volume of approximately 6 L. Details for the DMS analytical system and the DMS measurement protocol are described in Jang et al (2016).

The distribution of aerosol particle sizes was measured at the Gruvebadet observatory, which is approximately 1 km southwest of Ny-Ålesund and approximately 60 m.a.s.l.  Two discrete systems of scanning mobility particle sizer (SMPS) systems, each of which included a differential mobility analyzer (DMA) and a condensation particle counter (CPC), continuously measured the distribution of small particles in in differential mobility equivalent diameter ranges 3–60 nm (combination of TSI 3085 and TSI 3776) and 10–500 nm (TSI 3034), and an aerodynamic particle sizer (APS) analyzed larger particles in the range 0.5–20 µm in diameter (Park et al., 2014; Lupi et al., 2016).

A high volume air sampler equipped with a PM$_{2.5}$ impactor (collecting particles < 2.5 µm in aerodynamic equivalent diameter) was used for collection of aerosol samples. The sampler was mounted on the roof of the Gruvebadet observatory, and sampled particles every 3 days between 9 April and 20 May 2015, and later measured concentrations of major ions and the stable S isotope composition on a quartz filter.

For measurement of major ions (Na$^+$, K$^+$, Mg$^{2+}$, SO$_4^{2-}$, Cl$^-$, and MSA), a 47-mm (diameter) disk filter was punched out from a PM$_{2.5}$ aerosol quartz filter. All major ions collected on the disk filter were extracted in 50 mL Milli-Q water and analyzed by a Dionex ion chromatography system (Thermo Fisher Scientific Inc., USA). The concentrations of major anions were determined using a Dionex model ICS-2000 with an IonPac AS 15 column and the concentrations of major cations were determined using a Dionex model ICS-2100 with an IonPac CS 12A column. Three times the standard deviations of blank measurements were used as detection limits (0.01 to 0.26 ng mL$^{-1}$) (Kang et al., 2015).

For measurement of the stable S isotope ratio ($^{34}$S/$^{32}$S), all S compounds on half of a PM$_{2.5}$ quartz filter were extracted in 50 mL Milli-Q water. The filtrate was treated with 50–100 µL of 1M HCl to adjust the solution to pH = 3–4. Then, 100 µL of 1M BaCl$_2$ was added to cause all S as SO$_4^{2-}$ to precipitate as BaSO$_4$. After a 24-h precipitation period at room temperature, the BaSO$_4$ precipitate was recovered by filtration on a membrane filter and finally dried for 24 h. Each membrane filter containing BaSO$_4$ was packed into a tin cup and analyzed by isotope ratio mass spectrometer (IsoPrime100, IsoPrime Ltd, UK) coupled to an elemental analyzer (Vario EL, Elementar Co, German). The resulting S isotope ratio of a sample ($\delta^{34}$S) was expressed as parts per thousand (‰) relative to the $^{34}$S/$^{32}$S ratio in a standard (Vienna Cañon Diablo Troilite) (Krouse and Grinenko, 1991).

$\delta^{34}$S (‰) = {($^{34}$S/$^{32}$S) $_{sample}$ / ($^{34}$S/$^{32}$S) $_{standard}$ − 1} X 1000
(1)

Information about the S isotope ratio of aerosol particles and the concentrations of major ions enabled the estimation of the relative contributions of biogenic DMS (f$_{bio}$), anthropogenic SO$_X$ (f$_{anth}$), and sea-salt SO$_4^{2-}$ (f$_{ss}$) to the total aerosol SO$_4^{2-}$ concentration. The concentration of ss-SO$_4^{2-}$ was first calculated by multiplying the Na$^+$ concentration (as a sea spray marker) by 0.252 (the seawater ratio of SO$_4^{2-}$/Na$^+$) (Keene et al., 1986). The nss-SO$_4^{2-}$ fraction of the total SO$_4^{2-}$ was then calculated by subtracting the fraction of ss-SO$_4^{2-}$ from the total SO$_4^{2-}$. Finally, the fraction of biogenic SO$_4^{2-}$ was estimated by solving the following equations:

$\delta^{34}$S$_{measured}$ = $\delta^{34}$S$_{bio}$ f$_{bio}$ + $\delta^{34}$S$_{anth}$ f$_{anth}$ + $\delta^{34}$S$_{ss}$ f$_{ss}$
(2)
f$_{bio}$ + f$_{anth}$ + f$_{ss}$ = 1
(3)
f$_{ss}$ = 0.252 Na$^+$/total SO$_4^{2-}$
(4)

In solving Eq. (2)–(4), the published S isotope end-member values of DMS-derived SO$_4^{2-}$ ($\delta^{34}$S$_{bio}$ = 18 ± 2‰), anthropogenic SO$_4^{2-}$ ($\delta^{34}$S$_{anth}$ = 5 ± 1‰), and sea-salt SO$_4^{2-}$ ($\delta^{34}$S$_{ss}$ = 21.0 ± 0.1‰) were used (McArdle and Liss, 1995; Norman et al., 1999; Böttcher et al., 2007; Lin et al., 2012).

메모 포함[박17]: (response to referee #1 and #2)
Q8. Provide more information about the use of DMA-CPC: Two discrete SMPS systems were used to measure the distribution of aerosol particle sizes. We have added more detailed information and references.

메모 포함[박18]: (response to referee #1)
Q9. we have included a measure of uncertainty associated with the $\delta^{34}$S$_{ss}$ value (21.0 ± 0.1‰) for sea salts.

메모 포함[박19]: (response to referee #1)
Q9. We have added this reference.

**3 Results and Discussion**

**3.1 Atmospheric DMS and aerosol particles**

[revised manuscript text omitted]

~~Anthropogenic SO$_X$ from the continents that reaches Ny-Ålesund will have undergone alterations during long-range transport. For example, hydrophilic H$_2$SO$_4$ particles derived from anthropogenic SO$_X$ tend to grow into particles larger than 100 nm (accumulation mode), especially following partial or total neutralization by NH$_3$, which in turn yields hygroscopic compounds such as NH$_4$HSO$_4$ and (NH$_4$)$_2$SO$_4$. In contrast, the photochemical oxidation of DMS (yielding biogenic SO$_4^{2-}$) occurs locally and the resulting biogenic particles more tend to develop into small particles (nucleation mode, 3–10 nm; and Aitken mode, 10–100 nm). These established explanations are generally consistent with our observations.~~ The transition of aerosol microphysical properties from a distribution dominated by an accumulation mode (Arctic haze period) to a distribution dominated by

메모 포함[박20]: (response to referee #2)
Q6. As suggested, we have changed "3-fold higher" to "more than double".

메모 포함[박21]: (response to referee #1)
Q10. Provide more description about the relation between atmospheric DMS mixing ratios and MSA concentration: Local DMS production explained much of the variation in MSA, but did not account for all MSA variations. Therefore, we have added a short paragraph noting that long range transport of MSA to the site was a possibility.

메모 포함[박22]: (response to referee #2)
Q7. Our description of Figure 1 was misleading. In the revised manuscript we have newly added this sentence.

메모 포함[박23]: (response to referee #2)
Q7. Our description of Figure 1 was misleading. In the revised manuscript we have removed this sentence.

메모 포함[박24]: (response to referee #1)
Q10. We have added "and MSA (21.4 ng m$^{-3}$)".

메모 포함[박25]: (response to referee #1)
Q12. We have replaced "probably" to "possibly".

메모 포함[박26]: (response to referee #1)
Q13. We have cited O'Dowd et al. (2002) and Allan et al. (2015), and added a brief explanation.

메모 포함[박27]: (response to referee #2)
Q8. Is DMS only responsible for the formation of small particles? Figure 1a shows that approximately 45% of the variance in small particle formation was explained by DMS. Iodine may have contributed to explaining the remaining variance. We have added statement that iodine could be an important contributor.

메모 포함[박28]: (response to referee #2)
Q9. This study did not provide direct evidence: We agree that molecular-scale measurements of chemical species actually involved in nucleation processes are required to provide direct evidence for the DMS-derived aerosol formation events. Therefore, we have added a short paragraph indicating the limitation of our study in this regard.

nucleation and Aitken mode atmospheric particles (phytoplankton bloom period) was probably driven by the combination of three factors, including changes in air mass transport, incoming solar radiation and condensation sink processes (Tunved et al., 2004 and 2013). Specifically, the large accumulation mode particles outnumbered the small nucleation and Aitken mode particles during early spring (April) but the concentration of those large particles decreased rapidly from April to May, with particles smaller 100 nm becoming dominant in May (Fi g. 3 and S1). The high concentration of small particles (< 100 nm) during phytoplankton bloom period (May) constitutes compelling evidence for new particle formation derived from local DMS emission (Sharma et al., 2012; Tunved et al., 2013). 
[revised manuscript text omitted]
_4^{2-}$ during Arctic phytoplankton blooms. Concurrent measurements of a suite of parameters (DMS, satellite-derived phytoplankton biomass, concentration and chemical composition of particles) supported the assertion that oceanic emission of DMS significantly affects the properties of sub-micrometer particles in the Arctic atmosphere.~~ This study demonstrated the close association between an increase in DMS and increases in the total mass concentration of nss-$SO_4^{2-}$ and MSA during the period of Arctic

메모 포함[박38]: (response to referee #1)
Q20. Need to acknowledge potential contribution of anthropogenic $SO_4^{2-}$ to NPF events during the phytoplankton bloom period: Unfortunately, we could not accurately estimate the relative contribution of biogenic versus anthropogenic $SO_4^{2-}$ to the formation and growth of aerosol particles. However, recent field observations indicate that a considerable amount of $SO_4^{2-}$ in aerosol particles having a diameter < 0.49 µm is biogenic (> 63%), based on size-segregated sulfur isotope analysis in the Arctic atmosphere. Therefore, we have added a short paragraph indicating the limitations of our study, and cited Ghahremaninezhad et al. (2016) in support of our argument.

메모 포함[박39]: (response to referee #2)
Q10. Need more evidence (such as box model) to support strong statements regarding DMS-derived aerosol formation: As supporting evidence that DMS can substantially contribute to the fine-mode particle formation we observed in the Arctic atmosphere, we have cited the modeling work of Chang et al. (2011), who showed that atmospheric DMS mixing ratios > 100 pptv are sufficient to account for the formation of ultrafine particles, particularly when background particle concentrations are low (condensation sink < 7.0 m$^{-2}$). To support the confidence of our statement, we have added a short paragraph describing the results of Chang et al. (2011) and have added this paper to the revised list of references.

메모 포함[박40]: (response to referee #2)
We have removed "*compelling*".

메모 포함[박41]: (response to referee #2)
We have removed "significantly".

[revised manuscript text omitted]

메모 포함[박66]: (response to referee #1)
Q11. As suggested, we have added the time series for solar irradiance to Figure 1b.

메모 포함[박67]: (response to referee #2)
Q10. We have added the calculated condensation sink.

[Figure]

**Figure 2:** (a) The concentrations of nss-$SO_4^{2-}$ (blue line), and MSA (red line) at the Gruvebadet observatory in April and May 2015. (b) The relationship between nss-$SO_4^{2-}$ and MSA measured in April (blue circles; Arctic haze period) and in May (red circles; phytoplankton bloom period). The red solid line shown in Fig. 2b indicates the best fit.

[Figure]

**Figure 3:** (a) Concentration, and (b) surface area of aerosol particles, including nucleation mode (3–10 nm), Aitken mode (10–100 nm), accumulation mode (100 nm–1 µm) and coarse mode (1–19 µm) particles in April and May 2015.

[Figure]

**Figure 4:** Relation between the nss-$SO_4^{2-}$/MSA ratio and the MSA concentration ($PM_{2.5}$; April and May 2015; grey circles). The dashed line indicates the bio-$SO_4^{2-}$/MSA ratio measured at Ny-Ålesund in 2014 (Udisti et al., 2016).

메모 포함[박68]: (response to referee #1)
Q18. We have moved Figure S3 to the main text.

[Figure]

**Figure 4 5:** Biogenic (bio) $SO_4^{2-}$ estimated using MSA and S isotope. (a) Biogenic $SO_4^{2-}$ as a percentage of the total aerosol $SO_4^{2-}$ burden. Black bars: biogenic $SO_4^{2-}$ estimated using MSA; grey bars: biogenic $SO_4^{2-}$ estimated using stable S isotope. (b) The relationship between the concentrations of biogenic $SO_4^{2-}$ estimated using MSA, and the concentrations of biogenic $SO_4^{2-}$ estimated using stable S isotope. The black solid line represents the best fit.

[Figure]

**Figure 5 6:** Relationships between the concentration of biogenic (bio) $SO_4^{2-}$ and the concentrations of particles of various sizes in April and May. (a) Particles 3–10 nm, (b) particles 10–100 nm, and (c) particles 100 nm–1 µm. The black solid lines in Fig. 5a and 5b 6a and 6b indicate the best fit between biogenic $SO_4^{2-}$ and the particle concentration during April and May 2015. Blue and red circles in Fig. 5c 6c indicate data obtained in April (Arctic haze period) and May (phytoplankton bloom period), respectively. The red solid line in Fig. 5c 6c indicates the best fit between biogenic $SO_4^{2-}$ and particle concentration in May 2015, and the blue solid line in the inset of Fig. 5c 6c indicates the best fit between anthropogenic (anth) $SO_4^{2-}$ and particle concentration in April 2015.

[Figure]

**Figure 6 7:** Spectral plot of number size distribution (dN/dlogD$_p$) as a function of particle diameter (D$_p$, 10–500 nm) and year day during April and May 2015. The black line represents the concentration of biogenic SO$_4^{2-}$.